



# The value of satellite remote sensing soil moisture data and the DISPATCH algorithm in irrigation fields

Mireia Fontanet[1,2,3], Daniel Fernàndez-Garcia[2,3], Francesc Ferrer[1]

[1]LabFerrer, Cervera, 25200, Spain

[2]Department of Civil and Environmental Engineering, Universitat Politècnica de Catalunya (UPC), Barcelona, 08034, Spain
[3]Associated Unit: Hydrogeology Group (UPC-CSIC)

*Correspondence to*: Mireia Fontanet (mireia@lab-ferrer.com)

**Abstract.** Soil moisture measurements are needed in a large number of applications such as climate change, watershed water balance and irrigation management. One of the main characteristics of this property is that soil moisture is highly variable

with both space and time, hindering the estimation of a representative value. Deciding how to measure soil moisture before undertaking any type of study is therefore an important issue that needs to be addressed correctly. Nowadays, different kinds of methodologies exist for measuring soil moisture; Remote Sensing, soil moisture sensors or gravimetric measurements. This work is focused on how to measure soil moisture for irrigation scheduling, where soil moisture sensors are the main methodology for monitoring soil moisture. One of its disadvantages, however, is that soil moisture sensors measure a small

volume of soil, and do not take into account the existing variability in the field. In contrast, Remote Sensing techniques are able to estimate soil moisture with a low spatial resolution, and thus it is not possible to apply these estimations to agricultural applications. In order to solve this problem, different kinds of algorithms have been developed for downscaling these estimations from low to high resolution. The DISPATCH algorithm downscales soil moisture estimations from 40 km to 1 km resolution using SMOS satellite soil moisture, NDVI and LST from MODIS sensor estimations. In this work,

DISPATCH estimations are compared with soil moisture sensors and gravimetric measurements to validate the DISPATCH algorithm in two different hydrologic scenarios; 1) when wet conditions are maintained around the field for rainfall events, and 2) when it is local irrigation that maintains wet conditions. Results show that the DISPATCH algorithm is sensitive when soil moisture is homogenized during general rainfall events, but not when local irrigation generates occasional heterogeneity. In order to explain these different behaviours, we have examined the spatial variability scales of NDVI and

LST data, which are the variables involved in the downscaling process provided by the MODIS sensor. Sample variograms show that the spatial scales associated with the NDVI and LST properties are too large to represent the variations of the average water content at the site, and this could be a reason for why the DISPATCH algorithm is unable to detect soil moisture increments caused by local irrigation.



## 1 Introduction

Soil moisture measurements taken over different spatial and temporal scales are increasingly required in a wide range of environmental applications, which include crop yield forecasting (Holzman et al., 2014), irrigation planning (Vellidis et al., 2016), early warnings for floods and draughts (Koriche and Rientjes, 2016) and weather forecasting (Dillon et al., 2016).

This is mostly due to the fact that soil moisture controls the water and energy exchange between key environmental compartments (atmosphere and earth) and hydrological processes, such as precipitation, evaporation, infiltration, and run-off (Ochsner, 2013; Robock et al., 2000).

There are several applications in which soil moisture measurements have been shown to provide relevant information (Robock et al., 2000). For example, in environmental applications, soil moisture is typically used for defining the water

stress occurring in natural and human systems (Irmak et al., 2000) or for quantifying nitrate leaching and drainage quality (Clothier and Green, 1994). The use of soil moisture measurements can also improve weather forecasting, which is currently based on atmospheric moisture. Here, we highlight that soil moisture measurements from the root zone yields important information for field irrigation scheduling, determining to a great extent the intensity and frequency of irrigation needed for plant growth as a function of water availability (Blonquist et al., 2006; Jones, 2004; Campbell, 1982).

Soil moisture is highly variable in both space and time, mainly as a result of the spatial variability in soil properties (Hawley, 1983), topography (Burt and Butcher, 1985), land uses (Fu, 1994) and vegetation (Le Roux et al., 1995). As a result, soil moisture data exhibit a strong scale effect that can substantially affect the reliability of predictions depending on the method of measurement. For this reason, it is important to understand how to measure soil moisture for irrigation scheduling in a commercial field site.

Nowadays, available techniques for measuring or estimating soil moisture can provide data on a small or large scale. Gravimetric measurements (Gardner, 1986) estimate soil moisture by the difference between the natural and the dry weight of a given soil sample. They are used as a true value of soil moisture for sensor calibration (Starr and Paltineanu, 2002) or soil moisture validation studies (Bosch et al., 2006; Cosh et al., 2006). The main disadvantage of this method is that these measurements are time-consuming; users have to go to the field to collect soil samples and place them in the oven for a long

time. Soil moisture sensors such as Time Domain Reflectometry sensors (Clarke Topp and Reynolds, 1998; Schaap et al.,




2003; Topp et al., 1980) or capacitance sensors (Bogena et al., 2007; Dean et al., 1987) are capable of measuring soil

moisture continuously using a data logger, thereby enabling the final user to save time. Soil moisture sensors are especially

useful for studying processes on a small scale, but suffer from the fact that field data is typically scarce and provides an

incomplete picture of a large area (Western et al., 1998). Nevertheless, the use of soil moisture sensors is a common practice

5    for guiding irrigation scheduling in cropping field systems (Fares and Polyakov, 2006; Thompson et al., 2007; Vellidis et al.,

2008).

Remote Sensing, which is often based on passive microwave radiometry, can estimate soil moisture over large areas

(Jackson et al., 1996). In this case, soil moisture estimations refer to the Near Surface Soil Moisture (NSSM), which

represents the first 5 cm of the top soil profile. In recent years, Remote Sensing techniques have been improved and

10   diversified their estimation, making them an interesting tool for monitoring NSSM and other variables such as the

Normalized Difference Vegetation Index (NDVI) and Land Surface Temperature (LST). Different satellites exist that are

capable of estimating NSSM, one of which is the SMOS (Soil Moisture and Ocean Salinity) satellite launched in November

2009 (Kerr et al., 2001). It has global coverage and a revisit period of 3 days at the equator, giving two soil moisture

estimations, ascending overpass at 6:00 am and descending overpass at 6:00 pm local solar time. The SMOS satellite  is a

15   passive 2D interferometer operating at L band (1.4 GHz) (Kerr et al., 2010). The spatial resolution ranges from 35 to 55 km,

depending on the incident angle. Its goal is to retrieve NSSM with a target accuracy of a 0.04 $m^3/m^3$ (Kerr et al., 2012). The

SMOS NSSM have been validated on a regular basis since the beginning of its mission (Bitar et al., 2012; Delwart et al.,

2008) and it is considered suitable for hydro-climate applications (Lievens et al., 2015; Wanders et al., 2014).

The relatively large variability of soil moisture compared to the low resolution SMOS NSSM data hinders the direct

20   application of this method to irrigation scheduling. However, the need for estimating NSSM with a resolution higher than 35

– 55 km using Remote Sensing has increased for different reasons: 1) This data can be downloaded easily from different web

sites; 2) A field installation of soil moisture sensors is not necessary; and 3) No specific maintenance is needed. For these

reasons, in the last few years different algorithms have been developed to downscale Remote Sensing soil moisture data to

tens or hundreds of meters.



Chauhan et al., (2003) developed a Polynomial fitting method which estimates soil moisture at 25 km resolution. This method links soil moisture data with surface temperature, vegetation index and albedo. It does not require in situ measurements but cannot be used under cloud coverage conditions. The change in the detection method reported by Narayan et al. (2006) downscales soil moisture at 100 m resolution. This is an optimal resolution for agricultural applications, but the

method is highly dependent on the accuracy of its input data. The same problem is attributed to the Baseline algorithm for the SMAP (Soil Moisture Active Passive) satellite (Das and Mohanty, 2006), which downscales soil moisture at 9 km resolution. These algorithms have to be validated using in situ measurements. For this purpose, most authors use soil moisture sensors installed at the top soil profile, i.e., the first 5 cm of soil (Albergel et al., 2011; Cosh et al., 2004; Jackson et al., 2010), while others use gravimetric soil moisture measurements (Merlin et al., 2012) or the combination of both

methodologies (Robock et al., 2000).

The DISPATCH method (DISaggagregation based on Physical And THeorical Change) (Merlin et al., 2012; Merlin et al., 2008) is another algorithm that downscales SMOS NSSM data from 40 km (low resolution) to 1 km resolution (high resolution). This algorithm uses Terra and Aqua satellite data to estimate NDVI and LST twice a day using the MODIS (Moderate Resolution Imaging Spectroradiometer) sensor. These estimations have 1 Km resolution and can be conducted if

there is no cloud coverage. This downscaling process provides the final user with the possibility of estimating NSSM using Remote Sensing techniques at high resolution. The DISPATCH algorithm has been validated (Malbéteau et al., 2015; Merlin et al., 2012; Molero et al., 2016) in fairly large and homogeneous areas without irrigation, but not in complex settings with changing hydrologic systems such as those representing a local irrigation field.

In this work, we evaluate the value of Remote Sensing in agricultural irrigation scheduling by comparing in situ soil

moisture data obtained from gravimetric and soil moisture sensors, with soil moisture data determined by downscaling Remote Sensing information with the DISPATCH algorithm.

## 1.1 Study Area

The study area is located in the village of Foradada (1.015 lat, 41.866 lot), in the Segarra – Garrigues (SG) system (Lleida, Catalonia). The SG system is an important hydraulic project currently being carried out in the province of Lleida, Catalonia,

which involves converting most of the current dry land fields into irrigated fields. Its construction enables 1,000 new



hectares with a long agricultural tradition to be irrigated in much of the dry land. To achieve this, an 85 km-long channel was constructed to supply water for irrigation. At present, approximately 16,000 irrigators are potential beneficiaries of these installations. However, most farmers have yet to install this irrigation system, which means that the SG systems are still regarded as dry land.

5    The Urgell area is located in the west of the SG system. This area has totally different soil moisture conditions, especially during the summer season when the majority of fields are currently irrigated. This gives rise to two clearly distinguishable wet and dry soil moisture conditions. Figure 1 shows these areas using aerial photography.

Figure 2 shows the Foradada field, which represents 20 ha of a commercial field irrigated by a solid set sprinkle irrigation system distributed with 18 different irrigation sectors. The soil texture is 65.6% Clay, 17.6% Silt and 16.8 Sand. Every year

10   two different crops are grown, the first one during the winter and spring seasons, when wet conditions are maintained by precipitation, and the second during the summer and autumn seasons, when wet conditions are maintained by irrigation. The Foradada field is thus one of the few irrigated fields located within the SG system. Consequently, this field has soil moisture conditions similar to those in the surrounding area during the winter and spring season, but completely different conditions during the summer and autumn seasons. This makes this site unique for assessing Remote Sensing in a distinct isolated field.

## 2 Materials and Methods

### 2.1 In situ Soil Moisture Measurements

A total of 9 intensive and strategic field campaigns were conducted in the study area during 2016: DOY42, DOY85, DOY102, DOY187, DOY194, DOY200, DOY215, DOY221 and DOY224. During each field campaign, disturbed soil samples were collected at the top soil profile (0-5 cm depth) for measuring gravimetric soil moisture content. A total of 101

20   measurement points, depicted in Fig. 3., were defined around the field.  They are divided into two different kinds of points: 1) Cross section points; 75 points defined to represent the spatial variability of soil moisture in different cross sections. In these cross sections, points are separated by 9, 16 and 35 m; 2) Support points; 26 points complement information measured from cross sections, thereby adding and supporting information about field spatial variability.



Each soil sample is analyzed using the gravimetric method for measuring gravimetric soil moisture content, which is transformed to volumetric soil moisture content using bulk density measurements (Letelier, 1982). Soil moisture is also measured using capacitive EC-5 sensors (METER Group, Pullman, WA, USA), previously calibrated in the laboratory (Star and Paltineanu, 2002). As Figure 3 shows, a total of 5 control points are installed across one of the three gravimetric cross

sections. Each control point represents a different irrigation sector of the field. Soil moisture sensors are installed at 5 cm depth, taking into account the explore volume of these sensors. Their resolution is ±0.03 cm$^3$·cm$^{-3}$. They are connected to an EM50G dataloggers (METER Group, Pullman, WA, USA) that register soil moisture every 5 minutes.

### 2.2 Remote Sensing Soil Moisture Measurements

The main objective of the DISPATCH algorithm is to downscale NSSM data obtained from SMOS at 40 km resolution to 1

km resolution; the downscaling technique decouples soil evaporation from 0-5 cm soil layer and the vegetation transpiration from the root zone layer by separating MODIS surface temperature into soil (LST data) and vegetation components (NDVI data), as in the Soil Evaporative Efficiency (SEE).

The estimation of SEE is assumed to be approximately constant during the day, given clear sky conditions. This MODIS-derived SEE is further considered as a proxy for the NSSM variability within the SMOS pixel. A downscaling relationship

and the SEE model provide the link between NSSM and SEE. The downscaling relationship is given by Eq. (1):

$$\theta_{HR} = \theta_{SMOS} + \left(\frac{\partial \theta_{mod}}{\partial SEE}\right)(SEE_{HR} - SEE_{SMOS}), \tag{1}$$

where $\theta_{SMOS}$ is the SMOS soil moisture, $SEE_{HR}$ the MODIS-derived SEE at a 1 km resolution, $SEE_{SMOS}$ is an average within the SMOS pixel at a 40 km resolution, and $\frac{\partial \theta_{mod}}{\partial SEE}$ the soil moisture partial derivative with respect to the SEE evaluated at SMOS scale. The formula for estimating this partial derivative is given by Eq. (2) (Merlin et al., 2013):

$$SEE = \frac{\theta}{\theta_p}, \tag{2}$$

where $\theta$ represents the soil moisture in the $0 - 5$ cm soil layer and $\theta_p$ the empirical parameter depending on soil properties and atmospheric conditions.

In this work, the DISPATCH algorithm has been executed during period DOY36 and DOY298 to estimate NSSM at 1 km resolution at the top soil profile in the Foradada field.





DISPATCH provides a daily NSSM pixel map, defining a grid, where the SG system and the Urgell areas are included. Soil moisture estimations from the Foradada pixel have been extracted. In this pixel, 51.5% of the total area corresponds to the irrigated area (the Foradada and another field) and the remaining pixel corresponds to dry lands (Figure 4). In this part we assume that DISPATCH estimations are influenced by both these conditions, and for this reason DISPATCH estimations

represent both conditions.

## 3 Results

One of the main advantages of the experiment presented here is that Remote Sensing soil moisture data is evaluated in two different hydrologic scenarios during the same year. The first scenario represents a wet period subject to rainfall events without irrigation and crop growth conditions. This period transpired during the winter and spring season, i.e., from February

to June. The subsequent scenario considered dry climate conditions with sprinkler irrigation operating upon crop demand during the summer and autumn season, from June to October. In contrast to the rainfall events, sprinkler irrigation creates a local artificial rainfall event using several rotating sprinkler heads. The irrigation system operates sequentially across the different field zones. The comparisons of these two hydrologic regimes allow us to evaluate the effect of local sprinkler irrigation.

Figures 5 compares gravimetric and soil moisture sensor measurements with the DISPATCH soil moisture data obtained from Remote Sensing under wet conditions without irrigation. The green region in the figure displays the daily minimum and maximum values of soil moisture data obtained with sensors. Error bars in the gravimetric measurements represent the standard deviation produced by the average of all the measurements. However, the error bars in the DISPATCH data correspond to the standard deviation of the algorithm estimates arising from two daily SMOS estimations, with four MODIS

data (two at 6:00am and two more at 6:00 pm). One may observe that gravimetric and sensor measurements yield similar results, thus indicating that this type of data is accurate and performs consistently well. The soil moisture data estimated by the DISPATCH algorithm seems to consistently underestimate the true value. However, it is worth mentioning that the DISPATCH data seem to be sensitive and follow the rainfall patterns. Figure 5 also shows the same data as in the previous figure, but using relative data. It verifies perfectly that DISPATCH estimations are sensitive to the rainfalls, and for this





reason the DISPATCH algorithm is capable of detecting relative soil moisture increments produced by general rainfall events.

Figure 6 again compares gravimetric and sensor soil moisture measurements with DISPATCH soil moisture estimations, but in this case the comparison is made during the dry period when the Foradada field is irrigated and under wet conditions, while most of the region is under dry conditions. One may observe that the DISPATCH data are essentially not sensitive to irrigation even though they respond properly to sporadic small rainfall events. Furthermore, the relative soil moisture values in Fig. 6. do not show significant patterns. Thus, even though the Dispatch algorithm seems to respond properly to significant rainfall events, with some bias during both the summer and the winter time, irrigation operating on a small scale remains undetected. The DISPATCH data disregards irrigation and merely indicates the dry conditions existing in the area. We conclude that the DISPATCH estimation provides representative estimates of the soil moisture conditions existing in the Foradada region at a resolution lower than expected.

## 4 Discussion

In this section, we seek to answer the important question of why the DISPATCH soil moisture estimations obtained by downscaling satellite information from 40 km to 1 km of resolution are not sensitive to irrigation. The DISPATCH resolution of 1 Km is similar to the characteristic scale of the Foradada irrigation field site and therefore a better performance was expected. To provide insight into this problem, we have examined the spatial variability scales of the different variables involved in the downscaling process, i.e., the NDVI and the LST properties provided by the MODIS satellite and used by DISPATCH to estimate SEE. In random field theory and geostatistics, the spatial variability and the estimation of the scale of variability is mainly characterized by the covariance function or by its equivalent, the semivariogram, which is defined by Eq. (3) (Journel and Huijbregts, 1978).

$$\gamma(\boldsymbol{h}) = \frac{1}{2} E\{[Z(\boldsymbol{x} + \boldsymbol{h}) - Z(\boldsymbol{x})]^2\}, \tag{3}$$

where $Z(\boldsymbol{x})$ is the random variable at the $\boldsymbol{x}$ position, and $E\{\cdot\}$ is the expectation operator. Essentially, the semivariogram is a function that measures the variability between pairs of variables separated by a distance $\boldsymbol{h}$. Very often, the correlation between two variables separated by a certain distance disappears when $|\boldsymbol{h}|$ becomes too large. At this instant, $\gamma(\boldsymbol{h})$



approaches a constant value. The distance beyond which $\gamma(\boldsymbol{h})$ can be considered to be a constant value is known as the range, which represents the transition of the variable to the state of negligible correlation. Thus, the range can ultimately be regarded as the size of independent objects. Since the spatial resolution of an image can be regarded as the size of independent bodies (Atkinson and Curran, 1997; Woodcock and Strahler, 1987), the range of variability in an image relates

directly to its resolution.

The NDVI and LST semivariograms were respectively estimated from the MOD13A2 and MOD11A1 product data, which can be freely downloaded from the Google Earth Engine website (https://earthengine.google.com). We selected a daily representative image of April, June and August. The April image represents a general rainfall event in the region, the June image shows when local irrigation starts in the Foradada field, and finally the August image represents when the crop is well

developed and frequent irrigation is needed. Figure 7 shows the LST and NDVI experimental semivariograms and fitted theoretical models, which have the following Eq. (4) and Eq. (5)

$$\gamma_{LST}(\boldsymbol{h}) = c_{11} \, \text{Sph}\left(\frac{|\boldsymbol{h}|}{a_{11}}\right) + c_{12} \left[1 - \cos\left(\frac{|\boldsymbol{h}|}{a_{12}}\,\pi\right)\right], \tag{4}$$

$$\gamma_{NDVI}(\boldsymbol{h}) = c_{21} \, \text{Exp}\left(\frac{|\boldsymbol{h}|}{a_{21}}\right) + c_{22} \, \text{Exp}\left(\frac{|\boldsymbol{h}|}{a_{22}}\right) + c_{23} \left[1 - \cos\left(\frac{|\boldsymbol{h}|}{a_{23}}\,\pi\right)\right], \tag{5}$$

where $c_{ij}$ are constant coefficients that represent the contribution of the different standard semivariogram models, and $a_{ij}$ are

the corresponding ranges of the different structures. Tables 1 and 2 present the semivariogram parameters adopted in the model. The analysis shows a nested structure with a positive linear combination between isotropic stationary standard semivariogram models (spherical and exponential models for the LST and NDVI, respectively) and the hole effect model. Hole effect structures most often indicate a form of periodicity  (Pyrcz and Deutsch, 2003). In our case, this periodicity reflects the presence of different areas with specific watering and crop growth characteristics. These different areas are the

Urgell area, which is based on irrigation, and the other is the SG area which is not and thus it is more difficult for the crop to grow.

The spatial variability of NDVI and LST vary with time according to the changes in hydrologic conditions. In April, the semivariogram of NDVI displays more variability and less spatial continuity, thereby reflecting the differences in growth rate and crop type existing on the regional scale during the wet season with rainfall events. On the other hand, the spatial

variability of LST is more significant in August. Importantly, results show that the scale of variability associated with the





MODIS characteristics during the dry season, when a controlled amount of water by irrigation is applied, ranges between 35 and 36 kms for the NDVI and between 22 and 32 kms for the LST. These spatial scales of variability are one order of magnitude larger than the MODIS resolution of 1 Km. In order to complement this, we have also determined the spatial scale of variability of the average water content $\theta_A$ associated with the Foradada field site, which is represented by the Eq. (6)

$\qquad \theta_A = \frac{1}{A} \int_A \theta(\boldsymbol{x}) d\boldsymbol{x}$, $\hfill$ (6)

where A is the support area and $\theta(\boldsymbol{x})$ is the point water cont soil moisture. If the spatial scale of an image is much larger than the spatial scale of $\theta_A$, the image will be unable to represent the fluctuations of $\theta_A$. The semivariogram of a local average stochastic process can be estimated by using the theory of regionalized variables, represented by Eq. (7) (Journel and Huijbregts, 1978), which determines that the regularized semivariogram is the average of the point semivariogram,

$\qquad \gamma_{A_1 A_2} = \frac{1}{A_1 A_2} \int_{A_1} \int_{A_2} \gamma(\boldsymbol{x} - \boldsymbol{y}) d\boldsymbol{x} d\boldsymbol{y}$, $\hfill$ (7)

where A is the support area and $\theta(\boldsymbol{x})$ is the point water cont soil moisture. If the spatial scale of an image is much larger than the spatial scale of $\theta_A$, the image will be unable to represent the fluctuations of $\theta_A$. The semivariogram of a local average stochastic process can be estimated by using the theory of regionalized variables (Journel and Huijbregts, 1978), which determines that the regularized semivariogram is the average of the point semivariogram by Eq. (8)

$\qquad \gamma_\theta(\boldsymbol{h}) = c \, \text{Exp}\left(\frac{|\boldsymbol{h}|}{a}\right)$, $\hfill$ (8)

The spatial scale of variability of $\theta(\boldsymbol{x})$ given by the range varies between 35 and 52 m. Substituting Eq. (8) into Eq. (7), we obtain that the regularized semivariogram of $\theta_A$ follows an exponential model with a range that varies between 200 and 500 m. One may thus see that the characteristic scale of $\theta_A$ needed to represent the soil moisture fluctuations in the Foradada field site is much smaller than the characteristic scale of the NDVI and LST properties given by MODIS. We therefore

conclude that the studied MODIS images cannot represent and distinguish the variability of $\theta_A$ in the Foradada field site, which might be the reason why the DISPATCH algorithm fails to describe the fluctuations in water content caused by irrigation. This is only noted during irrigation, because the corresponding spatial scale is given by the size of the given irrigation field. On the other hand, the typical spatial scale of rainfall events is much larger than this size, and therefore one





may see that the DISPATCH algorithm is capable of providing a better representation of these fluctuations during the wet season.

In sum, results indicate that caution should be paid when using datasets of soil moisture derived from satellite information and the DISPATCH downscaling algorithm. The mismatch between the spatial scales of NDVI and LST with MODIS can be

determinant in some irrigated areas depending on the scale of observation. In this work, we found that these discrepancies will be more pronounced in areas smaller than approximately 10 kms (about 1/3 of the range of LST). This seems to indicate that irrigation scheduling based on satellite information can be appropriate in other regions of the world with extensive irrigation surface coverage (e.g., Punjab basin). This can improve the efficiency of irrigation with little efforts. Yet, the ultimate performance will depend very much on the uniformity of the irrigation method, which is often diverse and

substantially varies in space and time.

## 5 Conclusions

We analyze the value of the Remote Sensing information (DISPATCH algorithm) for predicting soil moisture variations in a relatively small irrigation field site with a characteristic size of 20 Ha. The DISPATCH algorithm based on the NDVI and LST data obtained from the MODIS satellite is used for downscaling the SMOS information and transforming the SMOS

soil moisture estimations from a resolution of 40 Km to 1 km. These estimates are then compared with gravimetric and soil moisture sensors  measures taken on a small support scale over the field site.

The results show that the downscaled soil moisture estimations are capable of predicting the variations in soil moisture caused by rainfall events, but fail to predict those soil moisture estimates affected by irrigation at a local scale. To provide insight into this problem, we examine the spatial variability scales of the different variables involved, i.e., the NDVI, the

LST and the average soil moisture over the field site. These results clearly show that the characteristic spatial scale associated with the NDVI and LST properties is too large to represent adequately the variations of the average water content at the site. This effect is not so significant during rainfall events, because the typical spatial scale of rainfall events is much larger than the characteristic size of the irrigated field site.





From a different perspective, these results also suggest that irrigation scheduling based on satellite information and the DISPATCH downscaling algorithm can be appropriate in regions of the world with irrigated surface areas larger than approximately 10 kms. However, caution should be paid to the direct application of this method as its performance will strongly depend on the spatiotemporal variation of the irrigation within the area. These variations can generate occasional

heterogeneity leading to the failure of the soil moisture prediction method.

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

**Figures**





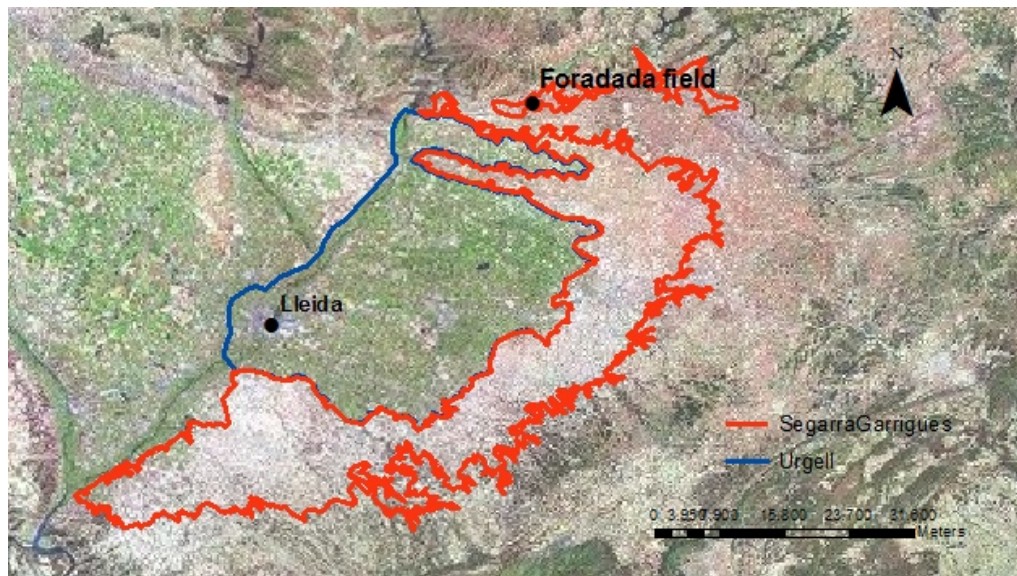

**Figure 1: The SG system, outlined in orange, in which dry land is observed, and the Urgell system, outlined in blue , in which most of the area is irrigated. The Foradada field is located to the north of the SG system, and the city of Lleida to the west of the Urgell system.**

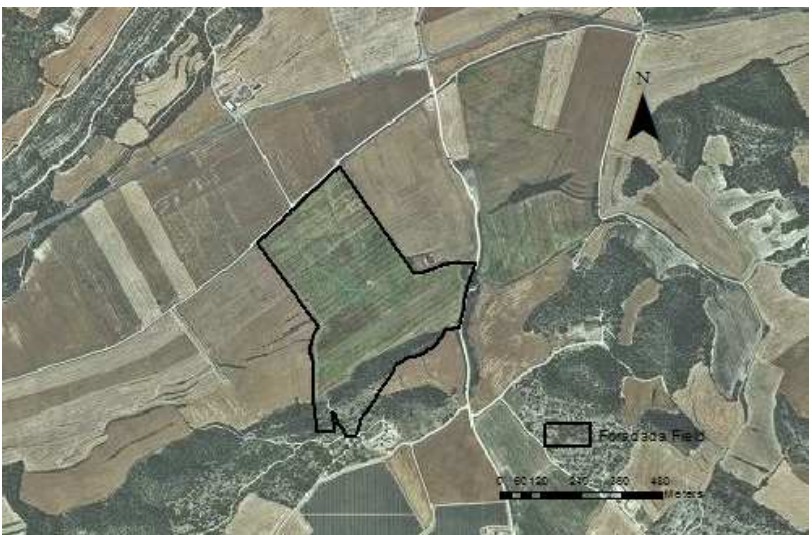



**Figure 2: The Foradada field, outlined in blue.**

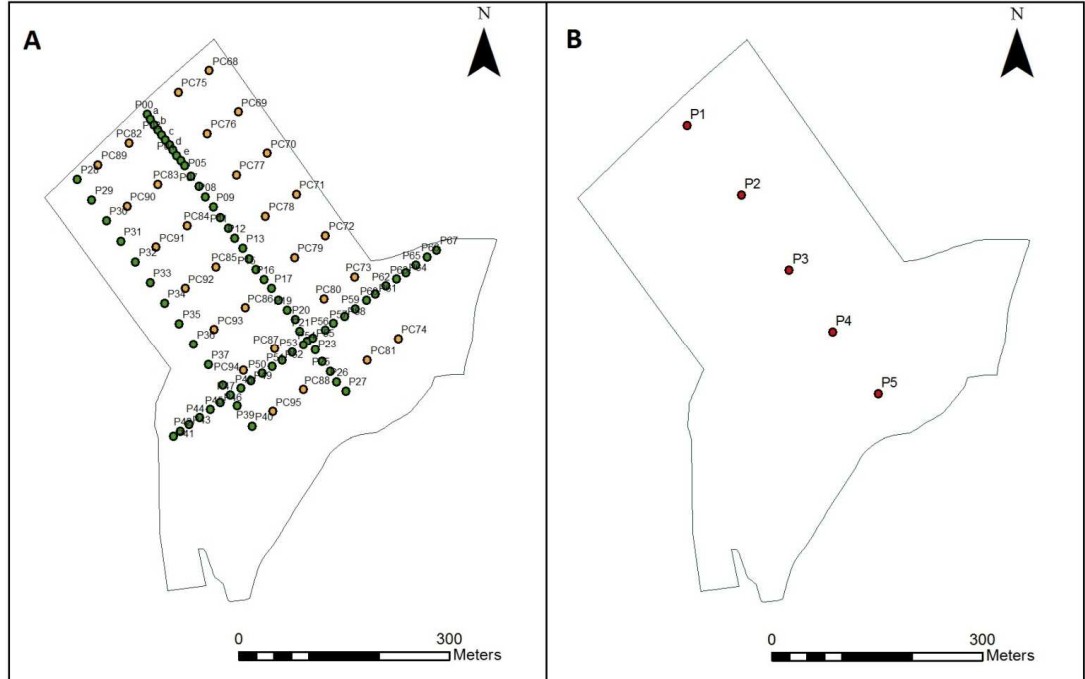

**Figure 3: A) Gravimetric measurement points, arranged with cross section points in green and support points in yellow. B) Control points, where EC-5 sensors were installed for measuring soil moisture content every 5 minutes.**



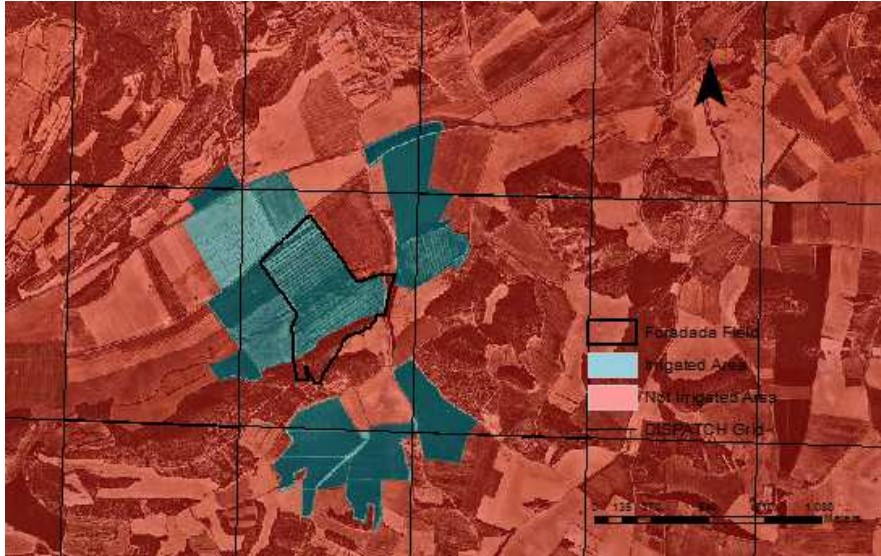

**Figure 4: The DISPATCH grid representing the Foradada field, outlined in dark blue; irrigated fields, in light blue; and dry land in light red.**





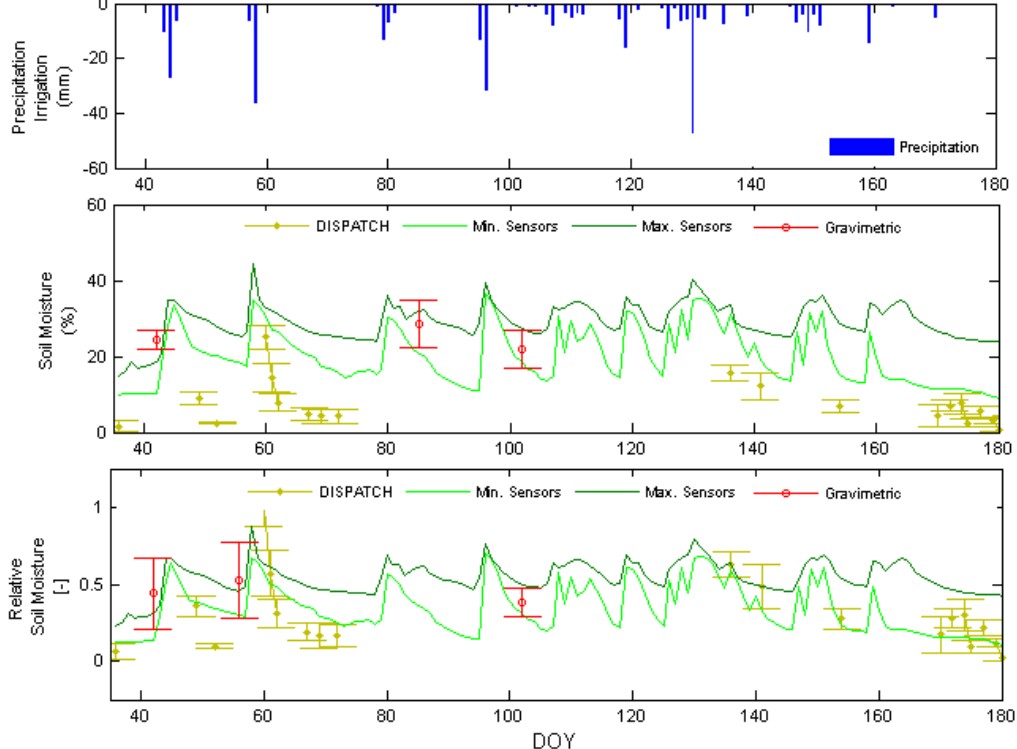

**Figure 5: Wet period representing rainfall events, marked in blue; gravimetric measurements, in red; daily maximum and minimum soil moisture sensors measurements, in green, and DISPATCH soil moisture estimations, in yellow: absolute values in the upper part and relatives values the lower part.**




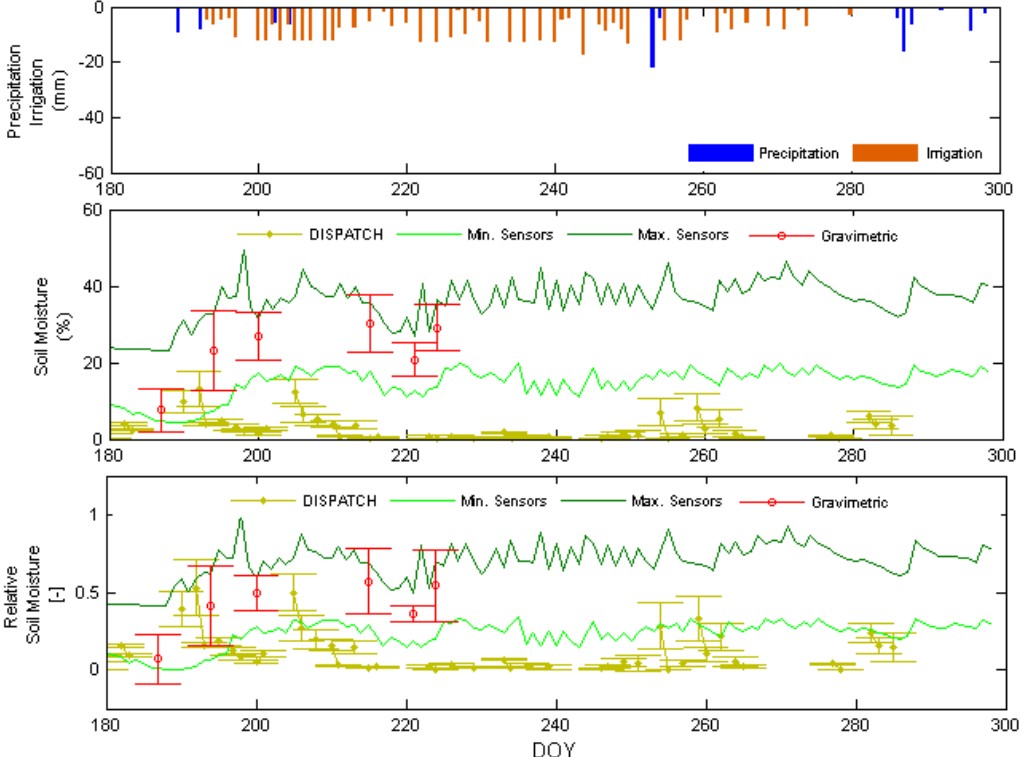

**Figure 6: Dry period representing local irrigation, marked in orange; sporadic rainfalls, in blue; gravimetric measurements, in red; daily maximum and minimum soil moisture sensors measurements, in green, and DISPATCH soil moisture estimations, in yellow: absolute values in the upper part, and relative values in the lower part.**





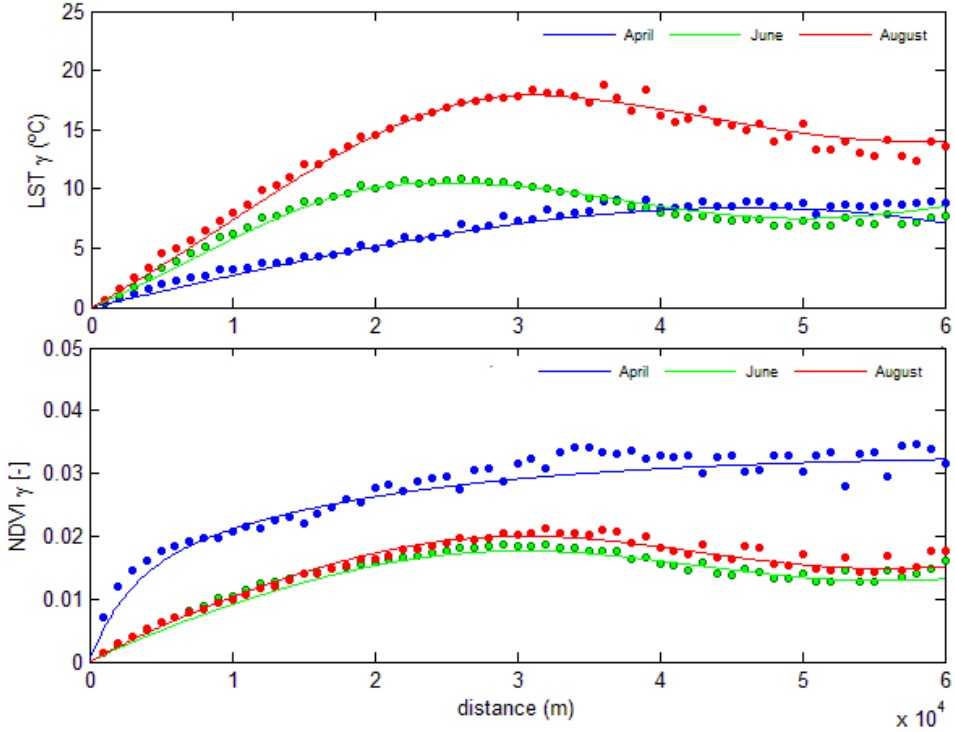

**Figure 7: LST and NDVI semivariograms showing 3 representative days in April, in blue; June, in green and August, in red.**





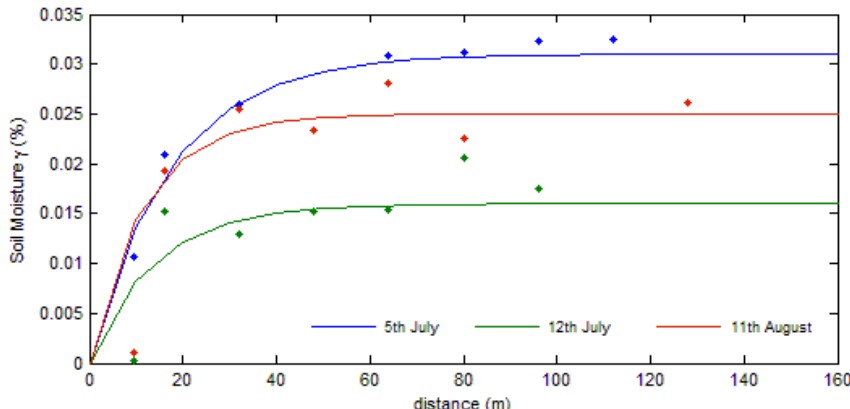

**Figure 8:** **Semivariograms of gravimetric measures from two representative days in July, shown in blue and green, and one day in August, in red.**

**Tables**

| LST | | | | | |
|---|---|---|---|---|---|
| | | Variogram | | Hole effect | |
| Month | Model | Sill ($c_{11}$) | Range ($a_{11}$) | Sill ($c_{12}$) | Range ($a_{12}$) |
| April | Spheric | 8.4 | 46000 | - | - |
| June | Spheric | 7.5 | 22000 | 1.5 | 25000 |
| August | Spheric | 14 | 32000 | 2 | 29000 |

5   Table 1. Fitting parameters of each LST semivariogram.

| NDVI | | | | | | | |
|---|---|---|---|---|---|---|---|
| | | Variogram | | | | Hole effect | |
| Month | Model | Sill ($c_{21}$) | Range ($a_{21}$) | Sill ($c_{22}$) | Range ($a_{22}$) | Sill ($c_{23}$) | Range ($a_{23}$) |
| April | Exponential | 0.013 | 8000 | 0.02 | 55000 | - | - |
| June | Exponential | 0.013 | 35000 | - | - | 0.22 | 28000 |
| August | Exponential | 0.015 | 36000 | - | - | 0.21 | 28000 |

Table2. Fitting parameters of each NDVI semivariogram.



| Gravimetric measurements | | | |
| --- | --- | --- | --- |
| | Variogram | | |
| Month | Model | Sill (c) | Range (a) |
| 5th July | Exponential | 0.031 | 52 |
| 12th July | Exponential | 0.016 | 42 |
| 11th August | Exponential | 0.025 | 35 |

Table 3. Fitting parameters of each Gravimetric measurements semivariogram.