# Peer review of "The value of satellite remote sensing soil moisture data and the DISPATCH algorithm in irrigation fields"

_Hydrology and Earth System Sciences, 2018_

## Referee Comment (RC1) · Anonymous Referee #1 · 26 May 2018

**OVERVIEW**

The manuscript investigates the capability of satellite soil moisture observations to detect irrigation in a small agricultural area in north eastern Spain. Specifically, the 1-km disaggregated soil moisture product obtained by using SMOS and MODIS sensors, through the DISPATCH algorithm, is considered. Results for one-year observations

are shown and also compared with in situ soil moisture observations.

**GENERAL COMMENTS**

The topic of the manuscript is surely of interest for the readership of HESS. The use of satellite soil moisture observations for detecting and quantifying irrigation water is an emerging topic and we have to investigate which product is performing better in reproducing the irrigation signal depending on its spatial and temporal resolution plus accuracy, and also by considering the irrigation practices, the climatic conditions, soil and land use, etc. The paper investigated this interesting topic with a well-defined

ground-based observations network including not only meteorological observations but also in situ soil moisture data and irrigation water observations (usually missing). The ground-based dataset is perfectly designed for the purpose of the paper. Therefore, I believe the paper might deserve to be published. However, at this stage, I found some important shortcoming that, in my opinion, must be addressed before the publication. I listed below the general comments with also their importance.

1) MAJOR: The text of the manuscript does not read well in many parts. In the specific comments, I added some suggestions for the abstract only. The whole text should be revised avoiding repetitions, improving English writing (but I am not mother-tongue), and taking care to write accurately symbols, equations, acronyms. Being a scientific paper, the structure and the methodology used should be clear to the readerships.

2) MAJOR: The authors found that the 1-km SMOS soil moisture product is not suitable to detect small scale irrigation, even though theoretically the 1-km resolution of the

product should be suitable for detecting irrigation in the investigated area. The authors investigated spatial variability of NDVI and LST and found it is much larger than soil moisture observations. However, the spatial extent of LST and NDVI is much larger (even if not specified in the text) than the extend of in situ soil moisture measurements, therefore the comparison should not be carried out. Moreover, the problem is not related to the spatial variability of NDVI or LST, but to their capability to detect the irrigation signal. Much better should be to carry out a specific analysis with NDVI and LST images to assess if they are able to "see" irrigation.

3) MAJOR: Related to point (2), I believe that the problem is the strong dependency of the disaggregated SMOS 1-km product to SMOS soil moisture product. SMOS has a spatial resolution of around 40 km, therefore it is not sensitive to small scale irrigation in the area. As the 1-km product is strongly dependent on SMOS, it is simply not suitable for detecting irrigation at field scale (we obtained similar results in scientific analyses we are doing). As mentioned above, the analysis of the NDVI and LST signal by MODIS should be carried out, even though the temporal resolution might be not good due to cloud coverage. I believe that if we want to consider a disaggregated soil moisture product for irrigation detection, a different strategy should be implemented.

4) MODERATE: As mentioned before, the text should be improved and specifically the structure of the paper. In Section 4 "Discussion" the theoretical background of geostatistical analysis is described. It should be moved to the methodology section.

**SPECIFIC COMMENTS**

Page 1, line 8: Soil moisture data are not really important for climate change studies.

Page 1, line 10: "with both space and time" is not correct, to be revised.

Page 1, line 12: Currently we can obtain soil moisture estimated through 1) in situ

observation (fixed stations and field measurements), 2) remote sensing (satellite, airplanes, drones), and 3) modelling (hydrological and/or climate). Please revise.

Page 1, line 13-14: "where soil moisture measurements . . .". Which measurements? Please revise.

Page 1, line 16: Currently we have Sentinel-1 that can provide 1-km soil moisture measurements. . .and also new techniques (e.g. CYGNSS).

Page 1, lines 19: Acronyms should be defined (SMOS, NDVI, LST, . . .).

Page 1, line 27: "reason for why" remove "for"

**RECOMMENDATION**

On this basis, I found the topic of the paper relevant and interesting but a major revision is required before the publication in Hydrology and Earth System Sciences.

─────────────────────────

---

## Referee Comment (RC2) · Anonymous Referee #2 · 9 Jul 2018

This manuscript aims to explore the value of the DisPATCh product to detect local irrigation. DisPATCh algorithm has been developed to downscale the SMOS data from 40 km to 1 km resolution using MODIS data (NDVI and LST). The approach has been assessed using an extensive ground measurements of surface soil moisture in the Foradada field in northern east of Spain during 2016. The large in situ data set makes part of the merits of this study. However, the authors provide light context and literature review to this research, and its importance. I feel the use of remotely sensed soil moisture to detect irrigation is a hot topic in semi-arid regions. For this reason, the topic of the manuscript is interesting for the readership of HESS. Nevertheless, I found some critical issues that must be addressed before the publication. Please find below

the general comments:

CRITICAL: The manuscript does not read well and it needs to be revised by improving the structure, avoiding repetitions, and by writing symbols, equations, acronyms consistently. Being a scientific paper, the structure has to be clear for the readerships. I found the introduction doesn't flow and lacks background information. Methods/Results/Discussion sections are confusing; methods are scattered throughout the sections and discussion reveals mainly results. Find comments and suggestions in the document attached.

CRITICAL: The authors investigated the spatial variability of NSSM, NDVI and LST. Although, the spatial resolution of LST and NDVI is 1 km (using MODIS dataset), the spatial resolution of soil moisture is few centimeters by using gravimetric measurements. Thus, the comparison does not make any sense and the respective discussion is wrong. It would be interesting to explore the value of DisPATCh and LST for different field scales over large areas, such as the SG region. Or you could explore LST and NDVI at high spatial resolution using Landsat data. Find comments and suggestions in the document attached.

MAJOR: The authors evaluated DisPATCh NSSM using in situ measurements, however this study needs to be fulfilled by a statistical analysis (Correlation, Bias etc . . . ). The result section would be improved by adding a temporal description/comparison of NSSM.

MAJOR: I don't think the concluding statement: "DisPACTh algorithm fails to describe the fluctuations in water content caused by irrigation" is correct; the current spatial resolution of DisPATCh might still be too coarse for local irrigation detection. However, DisPATCh succeeded to reveal spatial heterogeneities as rivers, irrigation areas, floods (Escorihuela et al. 2016, Malbeteau et al. 2015 2018, Molero et al 2016). It would be interesting to discuss the value and the limitation of DisPATCh over irrigated area (from local irrigation to large irrigation system). This conclusion needs to be balanced and

the limitation of the analysis performed in this study need to be considered.

MINOR: (1) Figures 1 to 4 need to be improved before publication. I suggest that they can be merged into one figure with two subfigures (figures 2, 3 and 5 into one map + zoom out figure 1 in order to see the coastline and Barcelona). (2) DisPATCh pixels on figure 4 are not squared, any explanation? is it really 1x1 km?

SPECIFIC COMMENTS: You will find comments in the document attached.

Please also note the supplement to this comment:
https://www.hydrol-earth-syst-sci-discuss.net/hess-2018-94/hess-2018-94-RC2-supplement.pdf

**Supplement:**

[revised manuscript text omitted]

---

## Author Comment (AC1) · 14 Jul 2018

Dear, We are grateful to you for the time and effort spent on the review of our manuscript. Our detail response and comments raised by you is attached. We believe our responses and the revisions made to the manuscript fully address the issues raised by the review. These revisions have helped clarify some aspects of our work and improve its interpretation.

Response to reviewer:

GENERAL COMMENTS

1) CRITICAL: The manuscript does not read well and it needs to be revised by improving structure, avoiding repetitions, and by writing symbols, equations, acronyms consistently. Being a scientific paper, the structure has to be clear for the readership. I found the introduction doesn't flow and lacks background information. Methods/Results/Discussion sections are confusing; methods are scattered throughout the sections and discussion reveals mainly results. Find comments and suggestions in the document attached.

- We have improved the manuscript taking into account your specific comments, avoiding repetitions, writing symbols, and equations consistently. You can see all the corrections in the specific comments section.

- We have also reorganized the manuscript to improve the structure and flow of the manuscript based on the comments raised by the two reviewers. In this context, we have added a sub-section entitled "Spatial resolution analysis" in section 3 (i.e., Materials and Methods). This way, the methods used to estimate the spatial resolution of variables (which where before introduced in the discussion section) were moved to the methods section. We hope this will largely improve the clarity of the manuscript.

2) CRITICAL: The authors investigated the spatial variability of NSSM, NDVI and LST. Although, the spatial resolution of LST and NDVI is 1 km (using MODIS dataset), the spatial resolution of soil moisture is few centimeters by using gravimetric measurements. Thus, the comparison does not make any sense and the respective discussion is wrong. It would be interesting to explore the value of DisPATCh and LST for different field scales over large areas, such as the SG region. Or you explore LST and NDVI at high spatial resolution using Landsat data. Find comments and suggestions in the document attached.

- There is some confusion here. The support volume of gravimetric soil moisture punctual measurements is few centimeters but the reviewer should notice that our comparison is not between point measurements and satellite information. The comparison

is between the average of these measurements over the entire field site (very well distributed with more than 100 measurement points) with satellite information. The average of the soil moisture is representative of the entire field site with a support volume of about 20 ha. Consequently, these two variables have similar support scale and therefore are comparable. We have rewritten part of the manuscript to clarify this issue.

- Another point along the same line is that soil moisture sensor data is also measured at the centimeter scale. This data is interesting because it shows the daily fluctuations of soil moisture. Sensors are well distributed over the entire field site but in this case we have only 5 sensors. Gravimetric measurements show that the average of soil moisture over the entire field site lays always between the maximum and minimum values of these sensors. Based on this, we have chosen to exhibit the minimum and maximum values of these 5 sensors in the figures. This way, the reader knows that the average soil moisture value lays within this region and can therefore appreciate the differences between the average soil moisture and satellite information in days where only sensor data is available. This point was also not clearly explained in the manuscript and we therefore understand the confusion of the reviewer. We have now rewritten the manuscript to clarify this point.

- We agree that it would be interesting to explore the value of DisPATCh and LST for different field scales over larger areas, such as the SG region, but the DISPATCH algorithm has been already well validated over large areas (Escorihulea et al. 2016, Malbeteau et al. 2015 2018, Molero et al. 2016) and we thought it is more interesting to analyze this under different conditions, i.e., punctual heterogeneity produced by local irrigation. Note that we already mentioned in the manuscript that the DISPATCH algorithm is capable to detect water bodies such as rivers, floods and large irrigated areas (page 11, line 7). MAJOR: The authors evaluated DisPATCh NSSM using in situ measurements, however this study needs to be fulfilled by a statistical analysis (Correlation, Bias etc..). The result section would be improved by adding a temporal description/comparison of NSSM.

- We sincerely do not understand this point, we have done more than this. We have conducted a geostatistical analysis of the key data involved, which is more than a simple statistical analysis. Even the field campaigns were designed to characterize the spatial variability. In the end, we decided to only show the variograms because we think it is the information needed to understand the discrepancy observed between satellite information and measurements. Moreover, the scope of the manuscript is not to report a geostatistical analysis but to understand the worth of satellite information for local irrigation.

MAJOR: I don't think that concluding statement: "DisPATCh algorithm fails to describe the fluctuations in water content caused by irrigation" is correct; the current spatial resolution of DisPATCh might still be too coarse for local irrigation detection. However, DisPATCh succeeded to reveal spatial heterogeneity as rivers, irrigation areas, floods 8Escorihuela et al. 2016, Malbeteau et al., 2015 2018, Molero et al., 2016). It would be interesting to discuss the value and the limitation of DisPATCh over irrigated area (from local to large irrigation system). This conclusion needs to be balanced and the limitation of the analysis performed in this study need to be considered.

- We have changed the sentence "DisPATCh algorithm fails to describe the fluctuations in water content caused by irrigation" with "DISPATCH algorithm did not properly repro­duce the temporal fluctuations of the average water content caused by local irrigation in this field site".

- To clarify the advantages of DisPATCh we have added in the introduction section: "DISPATCH succeed to reveal spatial heterogeneities as rivers, large irrigation areas and floods (Escorihulea et al. 2016, Malbeteau et al. 2015 2018, Molero et al. 2016).

MINOR: (1) Figures 1 to 4 need to be improved before publication. I suggest that they can be merged into one figure with two subfigures (figures 2, 3 and 4 into one map + zoon out figure 1 in order to see the coastline and Barcelona). (2) DisPATCh pixels on figure 4 are not squared, any explanation? Is it really 1x1 km?

- We think that we can merge Figure 1, 2 and 3 like the figure is shown below (Figure 1), but we think that merge also Figure 4 is too much information in a single figure.

- It is not exactly 1 x 1 km, it is 0.9 x 1.1 km.

SPECIFIC COMMENTS:

- Page 1 line 8: we delete "climate change ".

- Page 1 line 12: we can modify this part with: "Nowadays, different kinds of methodologies exist for measuring soil moisture; 1) in situ measurements, which can be obtained through fixed stations or field measurements, 2) Remote Sensing, where satellites, airplanes and drones estimate soil moisture, and 3) modeling, representing a hydrological system."

- Page 1 line 22: we can modify the sentence with "when irrigation maintains wet conditions".

- Page 1 line 27: we can delete "and"

- Page 2 line 12: we can delete "The use of soil moisture measurements can also improve weather forecasting, which is currently based on atmospheric moisture."

- Page 2 line 1: we can add some information: "Here, we highlight that soil moisture measurements from the root zone yields important information for field irrigation scheduling, determining to a great extent the duration and frequency of each irrigation needed for plant growth as a function of water availability (Blonquist et al., 2006; Jones, 2004; Campbell, 1982). Therefore, the main goal of irrigation scheduling is to apply the minimum volume of water guaranteeing maximum yield".

- Page 2 line 16: we can add " and with atmospheric conditions (Koster and Suarez, 2001)".

- Page 3 line 7: There is the possibility to remove: "which is often based on passive microwave radiometry".

- Page 3 line 25: we can connect better both sentences with "the first one is during the ascending overpass at 6:00 am and the second one is the descending overpass at 6:00 pm local solar time".

- Page 3 line 16: We can modify the sentence with "Since SMOS NSSM have been validated on a regular basis since the beginning of its mission (Bitar et al., 2012; Delwart et al., 2008), it is considered suitable for hydro-climate applications (Lievens et al., 2015; Wanders et al., 2014). "

- Page 4 line 7: We can delete authors and add "studies".

- Page 4 line 11: Your comment is "This makes it sound like its 'just another algorithm'. Rephrase the sentence in a way that introduces DISPATCH already as a superior method". We do not know or we do not have any reference that this algorithm is superior than the other algorithms.

- Page 4 line 17: Your comment is " Great! But why do we need it validated in irrigation fields? Highlight the importance of having this. Also, was there anywhere a mention between differences in soil moisture in irrigation vs rain fail? That is critical and missing here. We think that is necessary validate this algorithm in irrigation fields because one of the aim of this algorithm is monitor soil moisture for irrigation scheduling and management. Thus, this validation is the next step for the algorithm. We assume that precipitation and irrigation increase water content in the field and this process is measured by soil moisture sensors, but we consider that there is no difference between them except the scale effect (general rain fall versus local irrigation).

- Page 4 line 23: We can change "lot" by "lon".

- Page 5 line 5: We can change "has" by "represents"

- Page 6 line 8: We can change the title of the subsection " Remote Sensing Soil Moisture Measurements" by "DISPATCH Soil Moisture Measurements".

- Page 6 line 9: We can modify the sentence "The main objective of the DISPATCH

algorithm is to downscale" by "DISPATCH algorithm aims to downscale".

- Page 7 line 7: We can delete "Remote Sensing soil moisture".

- Page 10line 18: We can delete "One may thus see that".

- Page 11 line 14: We can change "information " by "NSSM".
* * *
[Figure]

**Fig. 1.** Figure 1

---

## Author Comment (AC2) · 16 Jul 2018

Dear,

We are grateful to the Editor and Reviewers for the time and effort spent on the review of our manuscript. We provide a detail response to your comments in this document. We believe that our responses and the revisions made to the manuscript fully address the issues raised by the reviewers. This revision has helped us to clarify some aspects of our work. Consequently, the manuscript has been largely improved.

Sincerely,

[Figure]

Mireia Fontanet on behalf of all co-authors.

Response to reviewer:

GENERAL COMMENTS

1) MAJOR: The text of the manuscript does not read well in many parts. In the specific comments, I added some suggestions for the abstract only. The whole text should be revised avoiding repetitions, improving English writing (but I am not mother-tongue), and taking care to write accurately symbols, equations, acronyms. Being a scientific paper, the structure and the methodology used should be clear to the readerships.

We have completely revised the manuscript to avoid repetitions, clarify some parts of the manuscript and improve English quality.

We have also modified mistakes regarding acronyms and symbols.

Specific comments have been corrected. Please, see the list of specific comments at the end of this document.

2) MAJOR: The authors found that 1-km SMOS soil moisture product is not suitable to detect small scale irrigation, even though theorically the 1-km resolution of the product should be suitable for detecting irrigation in the investigated area. The authors investigated spatial variability of NDVI and LST and found it is much larger (even if not specified in the text) than the extend of in situ soil moisture measurements, therefore the comparison should not be carried out. Moreover, the problem is not related to the spatial variability of NDVI or LST, but to their capability to detect the irrigation signal. Much better should be to carry put a specific analysis with NDVI and LST to assess if they are able to "see" irrigation.

The range of a semivariogram is the distance at which spatial correlation vanishes. This geostatistical property is used here to measure the size of independent image details. This is described at page 9, lines 2-5: "..., the range can ultimately be regarded as the size of independent objects. Since the spatial resolution of an image can be re-

garded as the size of independent bodies (Atkinson and Curran, 1997; Woodcock and Strahler, 1987), the range of variability in an image relates directly to its resolution". Of course, if the size of independent information content is too large compared to our field site, the satellite image cannot capture the spatial variation occurring at the scale of the field site. This is essentially the same as saying that there is no statistical difference between neighbor pixels. To further demonstrate this point, we can complement the geostatistical analysis with a visual comparison of the NDVI and LST pixel data obtained at a certain distance away from the Foradada pixel.

To do this, we have downloaded the neighbor NDVI and LST pixel data using MOD13A2 and MOD11A1 products with Google Earth Engine website from DOY036 to DOY298 (the same period of time as the execution of the DISPATCH algorithm). Two different pixel values have been extracted for each data set; 1) The Foradada field pixel value, where high values of NDVI and low values of LST are expected during irrigation period; 2) and the variable values of a pixel located 2 km away from the Foradada pixel value in the North-West direction. This pixel represents a dry land with expected lower NDVI and higher LST values compared to Foradada.

Figure 1, shows the temporal evolution of NDVI in both pixels. It can be observed that, during large-scale precipitation, both pixels represent similar conditions as expected, but there is no significant difference between them during local irrigation. If these periods of time are analyzed in detail, spring months, from DOY051 to DOY151, NDVI data from both pixels indicate that crop is growing as a consequence of a general precipitation that affects the full region, while during DOY152 to DOY298, NDVI values decreases because, on the other hand, in North-West pixel there is no irrigation, and on the other hand, the Foradada pixel NDVI data does not detect how crops develop as a consequence of irrigation.

The same case is shown with Fig.2., that represents LST temporal evolution in both pixels. In this case, LST also shows the same dynamics in both pixels even when irrigation is applied. As mentioned before, LST values form Foradada pixel should be

lower than North-West pixel if local irrigation was detected.

In our opinion, we think that this extra information clarifies the information in the manuscript.

3) MAJOR: Related to point (2), I believe that the problem is the strong dependency of the disggregated SMOS 1-km product to SMOS soil moisture product. SMOS has a spatial resolution of around 40 km, therefore it is not sensitive to small scale irrigation in the area. As the 1-km product is strongly dependent on SMOS, it is simply not suitable for detecting irrigation at a field scale (we obtain similar results in scientific analyses we doing). As mentioned above, the analysis of the NDVI and LST signal by MODIS should be carried out, even though the temporal resolution might be not good due to cloud coverage. I believe that if we want to consider a disaggregated soil moisture product for irrigation detection, a different strategy should be implemented.

We agree with your comment when you say that a different strategy for measuring soil moisture should be implemented, in fact, this is a part of our conclusions in page 11, line 17-18: The results show that the downscaled soil moisture estimations are capable of predicting the variations in soil moisture caused by rainfall events, but fail to predict those soil moisture estimates affected by irrigation at a local scale.

We would like to note that the DISPATCH method uses NDVI and LST information from Terra and Aqua satellite data to downscale soil moisture. The NDVI and LST satellite data is supposed to have a spatial resolution of 1 Km and therefore one should expect these estimates to be affected by local irrigation at the scale of the given field site (and consequently the DISPATCH product). The point here is that we actually see that the DISPATCH product is not affected, which calls for a reanalysis of the spatial resolution of these input variables. In doing this, we do not investigate whether the DISPATCH algorithm is adequate or not (we do not enter into the DISPATCH conceptualization) but we demonstrate that the DISPATCH input variables have smaller resolution than originally postulated. This is just one possible explanation among many others for the

failure of the DISPATCH algorithm in this case. In the revised manuscript, we will make this discussion clear.

4) MODERATE: As mentioned before, the text should be improved and specifically the structure of the paper. In section 4 "Discussion" the theoretical background of geostatistical analysis is described. It should be moved to the methodology section.

We have also reorganized the manuscript to improve the structure and flow of the manuscript based on the comments raised by the two reviewers. In this context, we have added a sub-section entitled "Spatial resolution analysis" in section 3 (i.e., Materials and Methods). This way, the methods used to estimate the spatial resolution of variables (which where before introduced in the discussion section) were moved to the methods section. We hope this will largely improve the clarity of the manuscript. We have also improved the manuscript in several editing aspects based on the comments raised by the two reviewer: avoiding repetitions, writing symbols, and equations consistently, improve English grammar and clarify confusing aspects about resolution and the use of scales.

SPECIFIC COMMENTS All specific comments that we agree, can be changed at the final version of the manuscript.

Page 1, line 8: Soil moisture data are not really important for climate change studies.

We have deleted "climate change studies" and added "hydro-climate approaches". Soil moisture measurements are needed in a large number of applications such hydro-climate approaches, watershed water balance and irrigation management.

Page 1, line 10: "with both space and time" is not correct, to be revised.

We have deleted "with both" and we have added "in". One of the main characteristics of this property is that soil moisture is highly variable in space and time, hindering the estimation of a representative value.

Page 1, line 12: Currently we can obtain soil moisture estimated through 1) in situ

observation (fixed stations and field measurements), 2) remote sensing (satellite, aire-planes, drones), and 3) modeling (hydrological and/or climate).

We have modified this sentence. Nowadays, different kinds of methodologies exist for measuring soil moisture; 1) in situ measurements, which can be obtained through fixed stations or field measurements, 2) Remote Sensing, where satellites, air-planes and drones estimate soil moisture, and 3) modeling, representing a hydrological system.

Page 1, line 13-14: "where soil moisture measurements. . ." Which measurements?

We have not found "where soil moisture measurements" in this line, anyway, in this line there is the sentence "where soil moisture sensors". In this case, we have not modified anything.

Page 1, line 16: Currently we have Setinel-1 that can provide 1-km soil moisture measurements. . . and also new techniques (e.g. CYGNSS)

Even though Setinel-1and other new techniques, such as CYNSS, provide soil mois-ture at 1 km resolution, we consider that it is not relevant information for abstract, but, we have added this information at the Introduction section: Other satellites, such as Sentinel-1, are able to estimate soil NSSM at 1 km resolution (Hornacek et al., 2012; Mattia et al., 2015; Paloscia et al., 2013). Sentinel-1 provides two kinds of products, the first one is Single Look Complex (SLC) and the second one is Ground Range Detected (GRD). The last one can be used in solving a wide range of problems of the Earth surface monitoring, such as soil moisture, but, it is not a direct measurement, thus data treatment is needed. In this case, GRD product is converted into radar backscatter coefficient and then into dB units to estimate soil moisture. Usually, these transforma-tions are not easy because these kind of measurements have surface roughness and vegetation influence on the signal (Garkusha et al., 2017; Wagner et al., 2010).

Page 1, line 19: Acronyms should be defined (SMOS, NDVI, LST. . .)

It is true and we have added acronyms definitions. The DISaggregation based on

Physical And Theorical CHange (DISPATCH) algorithm downscales soil moisture estimations from 40 km to 1 km resolution using Soil Moisture and Ocean Salinity (SMOS) satellite soil moisture, Normalized Difference Vegetation Index (NDVI) and Land Surface Temperature (LST) from Moderate Resolution Imaging Spectroradiometer (MODIS) sensor estimations.

Page 1, line 27: "reason for why" remove "for" We have deleted "for"

. . .the variations of the average water content at the site, and this could be a reason why the DISPATCH algorithm is unable to detect soil moisture increments caused by local irrigation.

[Figure]

**Fig. 1.** NDVI data from Foradada pixel and North-West pixel.

[Figure]

**Fig. 2.** LST data from Foradada pixel and North-West pixel.

---

## Author Response (AR1)

Dear Editor,

We are grateful to you, the Editor and the Reviewers for the time and effort spent on the review of our manuscript. Our detailed response to the comments raised by the Reviewers is attached. We believe our responses and the revisions made to the manuscript fully address the issues raised by the Reviewers. These revisions have helped clarify important aspects of our work and improve its presentation.

Page/Line numbers given in our response refer to the pages/lines of the ORIOGINAL and the NEW version of the manuscript to allow tracking the answer with respect to the original comments.

Sincerely,

Mireia Fontanet Ambròs on behalf of all co-authors

Legend

**Bold**: the comments and questions by the editor and the reviewers.

Blue: our answers.

Red: the detailed changes introduced in the manuscript.

Yellow: restructured

Green: rewritten

**Response to Reviewer 1:**

**GENERAL COMMENTS**
1) **MAJOR: The text of the manuscript does not read well in many parts. In the specific comments, I added some suggestions for the abstract only. The whole text should be revised avoiding repetitions, improving English writing (but I am not mother-tongue), and taking care to write accurately symbols, equations, acronyms. Being a scientific paper, the structure and the methodology used should be clear to the readerships.**

   - We have completely revised the manuscript to avoid repetitions, clarify some parts of the manuscript and also improve English quality.

   - We have also modified mistakes regarding acronyms and symbols, especially at the abstract.

   - Specific comments have been corrected. Please, see the list of specific comments at the end of this document.

2) **MAJOR: The authors found that 1-km SMOS soil moisture product is not suitable to detect small scale irrigation, even though theorically the 1-km resolution of the product should be suitable for detecting irrigation in the investigated area. The**

authors investigated spatial variability of NDVI and LST and found it is much larger (even if not specified in the text) than the extend of in situ soil moisture measurements, therefore the comparison should not be carried out. Moreover, the problem is not related to the spatial variability of NDVI or LST, but to their capability to detect the irrigation signal. Much better should be to carry put a specific analysis with NDVI and LST to assess if they are able to "see" irrigation.

- The range of a semivariogram is the distance at which spatial correlation vanishes. This geostatistical property is used here to measure the size of independent image details. This is described at page 8 line 4 (new version): *"The distance beyond which $\gamma(h)$ can be considered to be a constant value is known as the range, which represents the transition of the variable to the state of negligible correlation. Thus, the range can ultimately be seen as the size of independent objects in the image."*
  Of course, if the size of independent information content is too large compared to our field site, the satellite image cannot capture the spatial variation occur at the scale of the field site. This is essentially the same as saying that there is no statistical difference between neighbor pixels.
  To further demonstrate this point, we can complement the geostatistical analysis with a visual comparison of the NDVI and LST pixel data obtained at a certain distance away from the Foradada pixel.

  **Page 11 line 12 (new version): We have added this information**

  To further corroborate this point, Figure 6 compares the temporal evolution of LST and NDVI obtained from two adjoin MODIS pixels: the Foradada pixel and its North-West neighbour pixel. Note that the neighbor pixel corresponds to a not irrigated area. Data was downloaded using MOD13A2 and MOD11A1 products with Google Earth Engine website, from DOY036 to DOY298. In general, irrigation in an agriculture field site should produce a decrease in LST values and an increase in NDVI. However, Figure 6 shows the same dynamics and similar values in both pixels even when irrigation is applied. Results show that the LST and NDVI information cannot detect neither the sprinkler irrigation nor the crop growth as a consequence of irrigation in this case.

[Figure]

Figure 6. Temporal evolution of LST and NDVI obtained at the Foradada pixel and its neighbour North-West pixel situated 2 kms away.

In our opinion, we think that this extra information clarifies the information in the manuscript.

3) **MAJOR: Related to point (2), I believe that the problem is the strong dependency of the disggregated SMOS 1-km product to SMOS soil moisture product. SMOS has a spatial resolution of around 40 km, therefore it is not sensitive to small scale irrigation in the area. As the 1-km product is strongly dependent on SMOS, it is simply not suitable for detecting irrigation at a field scale (we obtain similar results in scientific analyses we doing). As mentioned above, the analysis of the NDVI and LST signal by MODIS should be carried out, even though the temporal resolution might be not good due to cloud coverage. I believe that if we want to consider a disaggregated soil moisture product for irrigation detection, a different strategy should be implemented.**

- We agree with your comment when you say that a different strategy for measuring soil moisture should be implemented, in fact, this is a part of our conclusions in page 12, line 7 (new version): *"From a different perspective, these results also suggest that irrigation scheduling based on satellite information coupled with the DISPATCH downscaling algorithm can be appropriate in regions of*

*the world with extensive irrigation surface coverage, larger than approximately 10 km (e.g., Punjab basin). However, caution should be paid to the direct application of this method as its performance will strongly depend on the spatiotemporal variation of the irrigation within the area. These variations can generate occasional heterogeneity leading to the failure of the soil moisture prediction method."*

We would like to note that the DISPATCH method uses NDVI and LST information from Terra and Aqua satellite data to downscale soil moisture. The NDVI and LST satellite data is supposed to have a spatial resolution of 1 km and therefore one should expect these estimates to be affected by local irrigation at the scale of the given field site (and consequently the DISPATCH product). The point here is that we actually see that the DISPATCH product is not affected, which calls for a reanalysis of the spatial resolution of these input variables. In the revised manuscript, we have added this information for making this discussion clear:

**We have added this information Page 10 line 2 (new version):**
We seek to answer the important question of why the DISPATCH soil moisture estimates obtained by downscaling satellite information from 40 km to 1 km of resolution are not sensitive to sprinkler irrigation in this case. The following possible sources of discrepancies can be identified: (i) errors associated with the approximations used in the DISPATCH downscaling formulation; (ii) differences in the scale of observations; and (iii) low quality of information associated with DISPATCH input variables. We concentrate the analysis on (ii) and (iii).

**MODERATE: As mentioned before, the text should be improved and specifically the structure of the paper. In section 4 "Discussion" the theoretical background of geostatistical analysis is described. It should be moved to the methodology section.**

- We have also reorganized the manuscript to improve the structure and flow of the manuscript based on the comments raised by the two reviewers. We have rewritten some parts of the manuscript. In this context, we have added a sub-section entitled "Spatial Resolution and Spatial Variability" in section 2 (i.e., Materials and Methods). This way, the methods used to estimate the spatial resolution of variables (which where before introduced in the discussion section) were moved to the methods section. The new manuscript structure is as follow:
  1. Introduction
     1.1 Study Area
  2. Materials and Methods
     2.1 In Situ Soil Moisture Measurements
     2.2 DISPATCH Soil Moisture Measurements
     2.3 Spatial Resolution and Spatial variability
  3. Results
     3.1 General Observations
     3.2 Analysis and Discussion
  4. Conclusions

We hope this will largely improve the clarity of the manuscript.

-   We have also improved the manuscript in several editing aspects based on the comments raised by the two reviewers: avoiding repetitions, writing symbols, and equations consistently, improve English grammar and clarify confusing aspects about resolution and the use of scales.

**SPECIFIC COMMENTS:**
Note that the main specific comments are not in the document because we have rewritten the manuscript.

-   **Page 1, line 8: Soil moisture data are not really important for climate change studies.**

    We have deleted "climate change studies" and added "hydro-climate approaches".

-   **Page 1, line 10: "with both space and time" is not correct, to be revised.**

    We have rewritten this part.

-   **Page 1, line 12: Currently we can obtain soil moisture estimated through 1) in situ observation (fixed stations and field measurements), 2) remote sensing (satellite, aire-planes, drones), and 3) modeling (hydrological and/or climate).**
    We have rewritten this part.

-   **Page 1, line 13-14: "where soil moisture measurements…" Which measurements?**
    We have rewritten this part.

-   **Page 4, line 16: Currently we have Setinel-1 that can provide 1-km soil moisture measurements… and also new techniques (e.g. CYGNSS)**

    Even though Setinel-1and other new techniques, such as CYNSS, provide soil moisture at 1 km resolution, we consider that it is not relevant information for abstract, but, we have added this information at the Introduction section (Page 4, line 6 new version):

    Other satellites, such as Sentinel-1, can estimate NSSM at 1 km resolution (Hornacek et al., 2012; Mattia et al., 2015; Paloscia et al., 2013). Sentinel-1 provides two kinds of products, the first one is Single Look Complex (SLC) and the second one is Ground Range Detected (GRD). The last one can be used for solving a wide range of problems related to Earth surface monitoring, such as soil moisture, but it is not a direct measurement and therefore data treatment is needed. In this case, GRD product is converted into radar backscatter coefficient and then into dB units to estimate soil moisture. Usually, these conversations are cumbersome because these kind of measurements have surface roughness and vegetation influence that affect the signal (Garkusha et al., 2017; Wagner et al., 2010).

-   **Page 1, line 19: Acronyms should be defined (SMOS, NDVI, LST…)**

    It is true and we have added acronyms definitions (Page 1, line 14, new version).

    … DISaggregation based on Physical And Theoretical scale CHange algorithm (DISPATCH) has been proposed in the literature to downscale soil moisture satellite data from 40 km to 1 km of resolution by combining the low resolution Soil Moisture Ocean Salinity (SMOS) satellite soil moisture data with the high resolution (Normalized

Difference Vegetation Index (NDVI) and Land Surface Temperature (LST) datasets obtained from a Moderate Resolution Imaging Spectroradiometer (MODIS) sensor.

- **Page 1, line27: "reason for why" remove "for"**
  We have rewritten this part.

**Response to Reviewer 2:**

**GENERAL COMMENTS:**

1) **CRITICAL: The manuscript does not read well and it needs to be revised by improving structure, avoiding repetitions, and by writing symbols, equations, acronyms consistently. Being a scientific paper, the structure has to be clear for the readership. I found the introduction doesn't flow and lacks background information. Methods/Results/Discussion sections are confusing; methods are scattered throughout the sections and discussion reveals mainly results. Find comments and suggestions in the document attached.**

   - We have improved the manuscript taking into account your specific comments, avoiding repetitions, writing symbols, and equations consistently. You can see all the corrections in the specific comments section.
   - We have rewritten the abstract:

**Abstract.** Soil moisture measurements are needed in a large number of applications such as hydro-climate approaches, watershed water balance management and irrigation scheduling. Nowadays, different kinds of methodologies exist for measuring soil moisture. Direct methods based on gravimetric sampling or Time Domain Reflectometry (TDR) techniques measure soil moisture in a small volume of soil at few particular locations. This typically gives a poor description of the soil moisture spatial distribution in relatively large agriculture fields. Remote sensing of soil moisture provides a large coverage and can overcome this problem but suffers from other problems stemming from its low spatial resolution. In this context, the DISaggregation based on Physical And Theoretical scale CHange algorithm (DISPATCH) has been proposed in the literature to downscale soil moisture satellite data from 40 km to 1 km of resolution by combining the low resolution Soil Moisture Ocean Salinity (SMOS) satellite soil moisture data with the high resolution (Normalized Difference Vegetation Index (NDVI) and Land Surface Temperature (LST) datasets obtained from a Moderate Resolution Imaging Spectroradiometer (MODIS) sensor. In this work, DISPATCH estimations are compared with soil moisture sensors and gravimetric measurements to validate the DISPATCH algorithm in an agricultural field during two different hydrologic scenarios; wet conditions driven by rainfall events and local sprinkler irrigation. Results show that the DISPATCH algorithm provides appropriate soil moisture estimates during general rainfall events but not when sprinkler irrigation generates occasional heterogeneity. In order to explain these differences, we have examined the spatial variability scales of NDVI and LST data, which are the input variables involved in the downscaling process. Sample

variograms show that the spatial scales associated with the NDVI and LST properties are too large to represent the variations of the average soil moisture at the site and this could be a reason why the DISPATCH algorithm is not working properly in this field site.

- We have also reorganized the manuscript to improve the structure and flow of the manuscript based on the comments raised by the two reviewers. In this context, we have added a sub-section entitled "Spatial resolution analysis" in section 3 (i.e., Materials and Methods). This way, the methods used to estimate the spatial resolution of variables (which where before introduced in the discussion section) were moved to the methods section. The new manuscript structure is as follow:
  1. Introduction
       1.1 Study Area
  2. Materials and Methods
       2.1 In Situ Soil Moisture Measurements
       2.2 DISPATCH Soil Moisture Measurements
       2.3 Spatial Resolution and Spatial variability
  3. Results
       3.1 General Observations
       3.2 Analysis and Discussion
  4. Conclusions

  We hope this will largely improve the clarity of the manuscript.

2) **CRITICAL: The authors investigated the spatial variability of NSSM, NDVI and LST. Although, the spatial resolution of LST and NDVI is 1 km (using MODIS dataset), the spatial resolution of soil moisture is few centimeters by using gravimetric measurements. Thus, the comparison does not make any sense and the respective discussion is wrong. It would be interesting to explore the value of DisPATCh and LST for different field scales over large areas, such as the SG region. Or you explore LST and NDVI at high spatial resolution using Landsat data. Find comments and suggestions in the document attached.**

- There is some confusion here. The support volume of gravimetric soil moisture punctual measurements is few centimeters but the reviewer should notice that our comparison is not between point measurements and satellite information. The comparison is between the averages of these measurements over the entire field site (very well distributed with more than 100 measurement points) with satellite information. The average of the soil moisture is representative of the entire field site with a support volume of about 25 ha. Consequently, these two variables have similar support scale and therefore are comparable.
  We have rewritten part of the manuscript to clarify this issue, Page 8, Line 23 (new version):
  Figures 3 compares gravimetric and soil moisture sensor measurements with the DISPATCH soil moisture estimates obtained from remote sensing data during the first period of time (without irrigation). We note that the comparison here is not between punctual gravimetric measurements (with support volume of few centimetres) and satellite information (1 km in resolution). We compare the

average of these punctual measurements over the entire field site (very well distributed with more than 100 measurement points) with satellite information. The average of the soil moisture is representative of the entire irrigated area associated with the Foradada field site. Consequently, these two variables have similar support scale and are therefore comparable. Error bars in the gravimetric measurements represent the standard deviation of all the measurements obtained in one day. In addition, the green region in this figure displays the daily minimum and maximum values of soil moisture data obtained from 5 EC-5 sensors.

- Another point along the same line is that soil moisture sensor data is also measured at the centimeter scale. This data is interesting because it shows the daily fluctuations of soil moisture. Sensors are well distributed over the entire field site but in this case we have only 5 sensors. Gravimetric measurements show that the average of soil moisture over the entire field site lays always between the maximum and minimum values of these sensors. Based on this, we have chosen to exhibit the minimum and maximum values of these 5 sensors in the figures. This way, the reader knows that the average soil moisture value lays within this region and can therefore appreciate the differences between the average soil moisture and satellite information in days where only sensor data is available. This point was also not clearly explained in the manuscript and we therefore understand the confusion of the reviewer.

We have now rewritten the manuscript to clarify this point. Page 9, Line 7 (new version):

We note that the average of gravimetric soil moisture data lays always within this region.

- We agree that it would be interesting to explore the value of DisPATCh and LST for different field scales over larger areas, such as the SG region, but the DISPATCH algorithm has been already well validated over large areas (Escorihulea et al. 2016, Malbeteau et al. 2015 2018, Molero et al. 2016) and we thought it is more interesting to analyze this under different conditions, i.e., punctual heterogeneity produced by local irrigation. Note that we already mentioned in the manuscript that the DISPATCH algorithm is capable to detect water bodies such as rivers, floods and large irrigated areas (page 4, line 19, new version).

3) **MAJOR: The authors evaluated DisPATCh NSSM using in situ measurements, however this study needs to be fulfilled by a statistical analysis (Correlation, Bias etc..). The result section would be improved by adding a temporal description/comparison of NSSM.**

- We sincerely do not understand this point; we have done more than this. We have conducted a geostatistical analysis of the key data involved, which is more than a simple statistical analysis. Even the field campaigns were designed to characterize the spatial variability. In the end, we decided to only show the variograms because we think it is the information needed to understand the discrepancy observed between satellite information and measurements. Moreover, the scope of the manuscript is not to report a geostatistical analysis but to understand the worth of satellite information for local irrigation.

4)  **MAJOR: I don't think that concluding statement: "DisPATCh algorithm fails to describe the fluctuations in water content caused by irrigation" is correct; the current spatial resolution of DisPATCh might still be too coarse for local irrigation detection. However, DisPATCh succeeded to reveal spatial heterogeneity as rivers, irrigation areas, floods (Escorihuela et al. 2016, Malbeteau et al., 2015 2018, Molero et al., 2016). It would be interesting to discuss the value and the limitation of DisPATCh over irrigated area (from local to large irrigation system). This conclusion needs to be balanced and the limitation of the analysis performed in this study need to be considered.**

-   We have changed the sentence "DisPATCh algorithm fails to describe the fluctuations in water content caused by irrigation" with Page 11, Line 25 (new version):
    Results have shown that in this case the downscaled soil moisture estimations are capable of predicting the variations in soil moisture caused by general rainfall events but fail to reproduce the temporal fluctuations of the average water content caused by local irrigation.

-   To clarify the advantages of DisPATCh we have added in the introduction section Page 4, Line 18 (new version):
    DISPATCH succeed to reveal spatial heterogeneities as rivers, large irrigation areas and floods (Escorihuela and Quintana-Seguí, 2016; Malbéteau et al., 2015, 2017; Molero et al., 2016) and it has also been validated (Malbéteau et al., 2015; Merlin et al., 2012; Molero et al., 2016) in fairly large and homogeneous irrigation areas, but not in complex settings with spatially changing hydrologic conditions such as those representing a local irrigation field.

**MINOR: (1) Figures 1 to 4 need to be improved before publication. I suggest that they can be merged into one figure with two subfigures (figures 2, 3 and 4 into one map + zoon out figure 1 in order to see the coastline and Barcelona). (2) DisPATCh pixels on figure 4 are not squared, any explanation? Is it really 1x1 km?**
We think that we can merge Figure 1, 2 and 3 like the figure is shown below (Figure 1), but we think that merge also Figure 4 is too much information in a single figure.

[Figure]

Figure 1. Location of the Foradada field site within the Segarra-Garriga irrigation system and distribution of soil moisture measurement points. Gravimetric measurement points are arranged with cross section points in green and support points in yellow. The location of EC-5 sensors are represented in red.

- It is not exactly 1 x 1 km, it is 0.9 x 1.1 km.

**SPECIFIC COMMENTS:**

Note that the main specific comments are not in the document because we have rewritten the manuscript.

- **Page 1 line 8:**
We have deleted "climate change"

- **Page 1 line 12:**

We have rewritten this part of the abstract.

- **Page 1 line 22:**
We have rewritten this part of the abstract.

- **Page 1 line 27:**
We have rewritten this part of the abstract.

- **Page 2 line 12:**
We have rewritten this part.

- **Page 2 line 1:**

We have added some information Page 2 line 7 (new version):

"Here, we highlight that soil moisture measurements from the root zone yields important information for field irrigation scheduling, determining to a great extent the duration and frequency of irrigation needed for plant growth as a function of water availability (Blonquist et al., 2006; Jones, 2004; Campbell, 1982)."

- **Page 2 line 16:**

we have added Page 2 line 11 (new version): " and with atmospheric conditions (Koster and Suarez, 2001)".

- **Page 3 line 7:**

We have rewritten this part.

- **Page 3 line 25:**

We have connected better both sentences with. Page 3, Line 8 (new version):

"It has global coverage and a revisit period of 3 days at the equator, giving two soil moisture estimations, the first one taken during the ascending overpass at 6:00 am and the second one during the descending overpass at 6:00 pm local solar time."

- **Page 3 line 16:**

We have modified the sentence with Page 3, Line 13 (new version):

"Since SMOS NSSM have been validated on a regular basis since the beginning of its mission (Bitar et al., 2012; Delwart et al., 2008), it is considered suitable for hydro-climate applications (Lievens et al., 2015; Wanders et al., 2014).

- **Page 4 line 7:**

We have deleted "authors" and added "studies". Page 4 line 2 (new version).

- **Page 4 line 11:**

Your comment is "This makes it sound like its 'just another algorithm'. Rephrase the sentence in a way that introduces DISPATCH already as a superior method".

We do not know or we do not have any reference that this algorithm is superior to the other algorithms.

- **Page 4 line 17:**

Your comment is "Great! But why do we need it validated in irrigation fields? Highlight the importance of having this. Also, was there anywhere a mention between differences in soil moisture in irrigation vs rain fail? That is critical and missing here.

We think that is necessary validate this algorithm in irrigation fields because one of the aim of this algorithm is monitor soil moisture for irrigation scheduling and management. Thus, this validation is the next step for the algorithm.

We assume that precipitation and irrigation increase water content in the field and this process is measured by soil moisture sensors, but we consider that there is no difference between them except the scale effect (general rain fall versus local irrigation).

- **Page 4 line 23:**

We have changed "lot" by "lon". Page 5 line 2 (new version).

- **Page 5 line 5:**

We have changed "has" by "represents". Page 5 line 11 (new version).

- **Page 6 line 8:**

We changed the title of the subsection " Remote Sensing Soil Moisture Measurements" by "DISPATCH Soil Moisture Measurements". Page 6 line 13 (new version).

- **Page 6 line 9:**

  We modified the sentence "The main objective of the DISPATCH algorithm is to downscale" by "DISPATCH algorithm aims to downscale". Page 6 line 15 (new version).

- Page 7 line 7:

  We have rewritten this part.

- **Page 10 line 18:**

  We have rewritten this part.

- **Page 11 line 14:**

  We have rewritten this part.

[revised manuscript text omitted]

---

## Author Response (AR2)

Dear Editor,

We are grateful to you and the Reviewer for the time and effort spent on the review of our manuscript. Our detailed response to the comments raised by you and the Reviewer is attached. These revisions have helped clarify aspects of our work and improve its presentation.

Page/Line numbers given in our response refer to the pages/lines of the ORIGINAL manuscript.

Sincerely,

Mireia Fontanet Ambròs on behalf of all co-authors

Legend

**Bold**: the comments and questions by the editor and the reviewers.

Blue: our answers.

Red: the detailed changes introduced in the manuscript.

**Response to Editor:**

**SPECIFIC COMMENTS:**

- **Page 1, Line 7:  change by "spatial distribution of soil moisture".**
  We changed it

- **Page 1, Line 25: change by "does not work".**
  We changed it.

- **Page 1, Line 29: change by "droughts".**
  We changed it.

- **Page 2, Line 14:**
  We dropped "in a commercial field".

- **Page 2, Line 15: add "either at" and "at a".**
  We added these words in the text.

- **Page 3, Line 8: change by "this satellite has"**
  We changed it.

- **Page 4, Line 9: change by "processing".**
  We changed it.

- **Page 4, Line 10: add "the".**
  We added it.

- **Page 4, Line 17: modify the sentence.**
  We modified with "conducted only if there is no cloud coverage."

- **Page 4, Line 18: modify by "success".**
  We modify it.

- **Page 4, Line 22: modify the word.**
  We modify it using value as according with the title of the manuscript.

- **Page 5, Line 3: change by "irrigation development project".**
  We changed it.

- **Page 5, Line 5: drop these words.**
  We dropped them.

- **Page 6, Line 2: add defined to.**
  We added these words.

- **Page 6, Line 3: add "about the spatial variability across the field".**
  We added these words.

- **Page 6, Line 11: modify by "registers".**
  We modify it.

- **Page 7, Line 20: add the "ground surface".**
  We added it.

- **Page 7, Line 20: drop "instead".**
  We dropped "Instead".

- **Page 8, Line 8: drop "finally".**
  We dropped it.

- **Page 8, Line 9: modify by "contend".**
  We modified it.

- **Page 8, Line 19:  How was this crop demand determined? Was this done using the satellite information - or through the sensor SM data?**
  The crop demand is not determined. This sentence has been added just for describing two different hydrologic scenarios; i) when there is precipitation, and ii) when irrigation satisfies crop water requirements. We can modify the "crop demand" by "crop water requirements".

- **Page 9, Line 1: change by "the point"**
  We changed it.

- **Page 9, Line 1: add "a support"**
  We added it.

- **Page 9, Line 2: add "instead"**

We added it.

- **Page 9, Line 2: change by "point".**
  We changed it.

- **Page 9, Line 3: drop "the".**
  We dropped it.

- **Page 9, Line 6: change by "the area between the light and dark green lines".**
  We changed it.

- **Page 9, Line 7: change by "the five EC-5 sensors".**
  We changed it.

- **Page 9, Line 7: change by "always lies".**
  We changed it.

- **Page 9, Line 8: modify the sentence.**
  We modified by" This supports the use of this information to complement".

- **Page 9, Line 8: add "on days".**
  We added it.

- **Page 9, Line 12: clarify how maximum and minimum soil moisture values have been determined:**
  We modified the sentence as:
   "…are the minimum and maximum values of the soil moisture time series data obtained with the EC-5 sensors."
  We hope that it clarifies how maximum and minimum soil moisture values have been determined.

- **Page 9, Line 13: drop "general".**
  We dropped it.

- **Page 9, Line 15: change by "underestimates"**
  We changed it.

- **Page 9, Line 19: change by "it can be seen that"**
  We changed it.

- **Page 9, Line 20: change by "even though there is a response to"**
  We changed it.

- **Page 9, Line 23: change by "it is not sensitive".**
  We changed it.

- **Page 10, Line 9: change by "contend" and "negligible".**
  We changed them.

- **Page 11, Line 10: correct by "1/10th".**
  We corrected it.

- **Page 11, Line 11: correct by "neighbouring pixels with a size of 1 km".**
  We corrected it.

- **Page 11, Line 12: corrected by "adjoining".**
  We corrected it.

- **Page 11, Line 13: clarify pixels compared.**
  Foradada area is assumed that is represented just for one pixel (1 km), see Page 9, Line 3: *"Average of the soil moisture is representative of the entire irrigated area associated with the Foradada field site. Consequently, these two variables have similar support scale and are therefore comparable. Error bars in the gravimetric measurements represent the standard deviation of all the measurements obtained in one day."* Thus in this section the main idea was compare two different pixels with theorically different hydrological scenarios and see if NDVI and LST were able to represent these different conditions. But the scale of these comparisons is the same.
  We modified the sentence with "*the Foradada pixel, where Foradada is located, and its North-West neighbour pixel.*"

- **Page 11, Line 13: change by "to an area that is not irrigated".**
  We change it.

- **Page 11, Line 17: clarify some aspects.**
  We tried to clarify some aspects of the discussion:

  We finally note that these results suggest that the resolution of LST and NDVI is not appropriate in this case but can also express that these two variables are simply not sensitive to irrigation because they only provide information about the status of the crop and land surface. Further research is needed in this sense.

- **Page 12, Line 11: clarify some aspects.**
  We change the word occasional heterogeneity by "different hydrological scenarios and behaviors".

- **Figure 1: improve Figure 1 (resolution and location).**
  We have improved resolution of Figure 1. Regarding the location, we added Barcelona coast because Segarra – Garrigues and Urgell areas are well appreciated in the figure. We think that if all Spain area is shown in the Figure, these areas will be too small in comparison with Spain area.

- **Figure 2: improve Figure 2 (legend).**
  We have improved Figure 2, modifying the legend.

**Response to Reviewer:**

**GENERAL COMMENTS**

1) **A quantitative comparison between in situ measurements and DISPATCH soil moisture is easy to be performed and would add quantitative results to the paper. For instance, the correlation between the average of in situ observations and DISPATCH in the two investigated periods will quantify the difference in the performance of DISPATCH product.**

- We agree that this information adds quantitative results to the paper that is why we have added two extra plots; the first one, represents the sensors average soil moisture measurements and DISPATCH data during rainfall periods, and the second one represents the same data but during local irrigation period. The information has been added at Page 10, Line 7 (new version).

This can also be seen from a different perspective by looking at the scatterplot between the average of the normalized relative soil moisture data obtained with the EC-5 sensors and the corresponding DISPATCH measurement determined at the same day. Figure 5 shows the scatterplots obtained during rainfall events and irrigation period. We note that even though a clear tendency is seen during rainfall events ($R^2$=0.57), no correlation seems to exist during irrigation ($R^2$=0.04).

[Figure]

**Figure 5. Scatterplot between the average of the normalized soil moisture obtained with EC-5 sensors and the DISPATCH measurements obtained during both hydrologic scenarios, rainfall events and irrigation period.**

We hope that this information will help and improve the data existing in the manuscript.

2) **The introduction contain the recent studies the attempted in using satellite soil moisture for providing irrigation information. For instance, see the papers by Lawston et al. (2017, doi: 10.1002/2017GL075733), Brocca et al. (2018, doi: 10.1016.j.jag.2018.08.023), Zaussinger et al. (2018, doi: 10.5194/hess-2018-388) and references therein.**

- We have added these references in the Introduction section improving the State of the Art. Page 4, Line 7 (new version):

Satellite soil moisture has been recently used for providing irrigation detection signals (Lawston et al., 2017), quantifying the amount of water applied (Brocca et al., 2018; Zaussinger et al., 2018), and estimating the water use (Zaussinger et al., 2018). All these works deal with relatively homogeneous and extensive irrigation surface coverages (several kms).

3) **The obtained results might be not only related to the spatial resolution of LST and NDVI, but also simply to the low sensitivity of these two variables to irrigation application. They provide information at the surface and of vegetation conditions, which might be not sensitive to irrigation. I am not aware of studies that have shown the LST or NDVI variation in time is sensitive to irrigation. I suggest adding this option should be added in the discussion of the results.**

- We have added this option in the discussion part Page 12, Line 6 (new version).

We finally note that these results suggest that the resolution of LST and NDVI is not appropriate in this case but can also express that these two variables are simply not sensitive to irrigation because they only provide information about the status of the crop and land surface. Further research is needed in this sense.

**SPECIFIC COMMENTS:**
- **Page 1, line 16: Remove the bracket before "Normalized"**

We deleted the bracket.

- **Page 1, Line 29: change "draught" with "drought".**
We changed the word.

- **Page 2, Line 1: Change "exchange" with "exchanges"**
We changed the word.

- **Page 2, Line 14: remove "in a commercial field site".**
We removed it.

- **Page 2, Line 17: change "true value" with "reference value".**
We changed the expression.

- **Page 2, Line 23: change "from the fact that field data is typically scarce and provides" with "from the typical low number of in situ sensors that provide".**
We changed the sentence.

- **Page 3, Line 3: remove comma after "Sensing". Why Remote sensing in capital.**
We removed comma and Remote Sensing capital letters.

- **Page 3, Line 4: add "(or less)" after 5 cm.**

We added.

- **Page 3, Line 3-14: The other satellites soil moisture products currently available should be mentioned, i.e., SMAP, ASCAT, AMSR2 and ESA CCI SM.**
  We added the follow information:

  Different satellites exist that are capable of estimating NSSM: the Soil Moisture Active Passive (SMAP) satellite, the Advanced Scatterometer (ASCAT) remote sensing instrument flown on board of the Meteorological Operational (METOP), the Advanced Microwave Scanning Radiometer 2 (AMSR2) instrument on board of the Global Change Observation Mission 1-Water (GCOM-W1) satellite, and the Soil Moisture and Ocean Salinity (SMOS) satellite launched in November 2009 (Kerr et al., 2001).

- **Page 3, Line 17: change "This data can be freely… web sites" with "data are freely available".**
  We changed it.

- **Page 3, Line 20: add a reference to the downscaling studies at the end of the sentence.**

  The manuscript includes different references for this sentence at the next paragraph where different downscaling algorithms are mentioned.

- **Page 3, Line 21: please, check, I do not think the spatial resolution in the study by Chauhan is 25 km.**
  Yes, you are right; the spatial resolution is 1 km. I have corrected the mistake.

- **Page 3, Line 23-24: the sentence "The change in… resolution" should be revised as it is not clear.**
  We modified the sentence with:
  The change in the detection method reported by Narayan et al. (2006) downscales soil moisture at 100 m resolution.

- **Page 4, Line 12: The recent study by Bauer-Marschallinger et al. (2018, doi: 10.1109/TGRD.2018.2858004) demonstrated the feasibility to obtain a soil moisture product operationally from Sentinel-1. I suggest adding this reference.**
  We added this reference.

- **Page 4, Line 16: change "even if" with "only if".**
  We changed it.

- **Page 5, Line 12: change "sprinkle" with "sprinkler".**
  We changed it.

- **Page 5, Line 12: please specify the source of the soil texture information. Does it refer to one point, a spatial average of multiple measurements?**
  We clarified the source of the information:
  The soil texture, in a single point, is 65.6% Clay, 17.6% Silt and 16.8 Sand.

- **Page 5, Line 23: remove the point after "Figure 1".**
  We removed the point.

- **Page 6, Line 10: change "resolution" with "accuracy" and specify how this number is obtained.**
  We changed the word.

- **Page 8, Line 9: what is the meaning of "content"? Also later in the text (P10, L9). Please revise.**
  We modified and correct the sentence.

- **Page 8, Line 10: add a reference here. Always, when you cite results of previous studies, a reference should be added.**
  We added the follow references:
  Atkinson and Curran, 1997; Curran, 1988

- **Page 8, Line 17: what is the meaning of "transpired"? Please revise.**
  We changed "transpired" by "occurs".

- **Page 9, Line 13: remove "general", also in the conclusions (P12, L1).**
  We removed them.

- **Page 9, Line 15: change "can slightly underestimate" with "slightly underestimates" (remove can).**
  We modified the word.

- **Page 9, Line 22: remove "significant".**
  We removed it.

- **Page 12, Line 8: change "can" with "might".**
  We change it.

[revised manuscript text omitted]

---

## Author Response (AR3)

Dear Editor,

We are grateful to you for the time and effort spent on the review of our manuscript. Our detailed response to the comments raised by you is attached. We have solved all the grammatical issues.

Page/Line numbers given in our response refer to the pages/lines of the ORIGINAL manuscript.

Sincerely,

Mireia Fontanet Ambròs on behalf of all co-authors

Legend

**Bold**: the comments and questions by the editor and the reviewers.

Blue: our answers.

Red: the detailed changes introduced in the manuscript.

**Response to Editor:**

**SPECIFIC COMMENTS:**

- **Page 1, Line 12:  add "of"**
  We added it.
- **Page 1, Line 15:  drop "of"**
  We dropped it.
- **Page 1, Line 20:  change by "and wet conditions driven by"**
  We changed it.
- **Page 3, Line 5:  drop "been"**
  We dropped it.
- **Page 3, Line 8:  drop "flown"**
  We dropped it.
- **Page 3, Line 9:  missing satellite word?**
  Yes, you are right. I added the word.
- **Page 4, Line 9:  drop "works"**
  We dropped it.
- **Page 4, Line 13:  missing a word?**
  Yes, you are right. I added the word.
- **Page 4, Line 13:  modify by "latter"**
  We modified it.
- **Page 4, Line 14:  modify by "conversions"**
  We modified it.
- **Page 4, Line 21:  modify by "have a resolution of 1 km"**
  We modified it.
- **Page 4, Line 22:  modify by "no cloud cover"**
  We modified it.
- **Page 4, Line 22:  modify by "such as"**
  We modified it.
- **Page 5, Line 2:  modify by "it has not been applied"**

We modified it.

- **Page 5, Line 9: modify by "dry-land"**
  We modified it.
- **Page 7, Line 8: modify by "which is defined in Eq. (2)"**
  We modified it.
- **Page 7, Line 18: modify by "has been applied during the period from DOY36 to DOY298 of 2016"**
  We modified it.
- **Page 8, Line 18: modify by "provide unnecessary data in others"**
  We modified it.
- **Page 8, Line 22: modify by "The first period represents wet soil conditions caused by natural rainfall"**
  We considered "crop growth" adds descriptive information and remarks that during this period a crop is growing in the field. We did not modify this sentence.
- **Page 8, Line 25: modify by "wet soil"**
  We modified it.
- **Page 9, Line 1: modify by "operating to satisfy crop water requirements"**
  We modified it.
- **Page 9, Line 3: remove "sole".**
  We removed it.
- **Page 9, Line 4: clarify the sentence.**
  We modified the sentence with " The comparisons of these two hydrologic periods allow us to evaluate the effect of local sprinkler irrigation on remote sensing soil moisture estimations."

- **Page 9, Line 9: add "the".**
  We added it.
- **Page 9, Line 14: add "the".**
  We added it.
- **Page 9, Line 18: add "and $\theta_{max}$".**
  We added it.
- **Page 9, Line 25: modify by "wet soil"**
  We modified it.
- **Page 10, Line 5: add "to"**
  We added it.
- **Page 11, Line 13: modify by "dry-land conditions in the SG area, the Urgell area is based on irrigation."**
  We modified it.
- **Page 12, Line 18: modify by "and LST"**
  We modified it.
- **Page 13, Line 1: drop "the".**
  We dropped it.

[revised manuscript text omitted]